# Best-case lower bounds in online learning

**Cristóbal Guzmán** [1,2]   **Nishant A. Mehta** [3]   **Ali Mortazavi** [3]

[1]Department of Applied Mathematics, University of Twente
[2] Institute for Mathematical and Computational Eng., Pontificia Universidad Católica de Chile
[3] Department of Computer Science, University of Victoria

`c.guzman@utwente.nl,nmehta@uvic.ca,alithemorty@gmail.com`

## Abstract

Much of the work in online learning focuses on the study of sublinear upper bounds on the regret. In this work, we initiate the study of best-case lower bounds in online convex optimization, wherein we bound the largest *improvement* an algorithm can obtain relative to the single best action in hindsight. This problem is motivated by the goal of better understanding the adaptivity of a learning algorithm. Another motivation comes from fairness: it is known that best-case lower bounds are instrumental in obtaining algorithms for decision-theoretic online learning (DTOL) that satisfy a notion of group fairness. Our contributions are a general method to provide best-case lower bounds in Follow the Regularized Leader (FTRL) algorithms with time-varying regularizers, which we use to show that best-case lower bounds are of the same order as existing upper regret bounds: this includes situations with a fixed learning rate, decreasing learning rates, timeless methods, and adaptive gradient methods. In stark contrast, we show that the linearized version of FTRL can attain negative linear regret. Finally, in DTOL with two experts and binary losses, we fully characterize the best-case sequences, which provides a finer understanding of the best-case lower bounds.

## 1   Introduction

A typical work in online learning would develop algorithms that provably achieve low regret for some family of problems, where low regret means that a learning algorithm's cumulative loss is not much larger than that of the best expert (or action) in hindsight. Such a work often focuses on algorithms that exhibit various forms of adaptivity, including *anytime* algorithms, which adapt to an unknown time horizon $T$; *timeless* algorithms, which obtain "first-order" regret bounds that replace dependence on the time horizon by the cumulative loss of the best expert; and algorithms like AdaGrad [8], which adapt to the geometry of the data. These examples of adaptivity all involve competing with the best expert in hindsight, but adaptivity comes in many guises. Another form of adaptivity involves upgrading the comparator itself: in the *shifting regret* (also known as the *tracking regret*) [11], the learning algorithm competes with the best sequence of experts that shifts, or switches, $k$ times for some small $k$. Naturally, an algorithm with low shifting regret can potentially perform much better than the single best expert in hindsight, thereby obtaining classical regret that is substantially negative.

Our work serves as a counterpoint to previous works: we show that a broad class of learning strategies provably fails, against *any* sequence of data, to substantially outperform the best expert in hindsight. Thus, these strategies are unable to obtain low shifting regret and, more generally, low regret against any comparator sequence that can be substantially better than the best expert in hindsight. More concretely, this paper initiates the study of *best-case lower bounds* in online convex optimization (OCO) for the general family of learning strategies known as Follow the Regularized Leader (FTRL) [1]. That is, we study the *minimum* possible regret of a learning algorithm over all possible sequences.

35th Conference on Neural Information Processing Systems (NeurIPS 2021).

As we will show, many instances of FTRL — including adaptive instances that are anytime, timeless, or adapt to gradients like AdaGrad — never have regret that is much less than the negation of the corresponding regret upper bounds. Thus, while these instances can be adaptive in some ways, they are in a sense prohibited from uniformly obtaining low regret for adaptive notions of regret like the shifting regret. For example, in the setting of decision-theoretic online learning (DTOL) with $d$ experts [9], the well-known anytime version of Hedge (which uses the time-varying learning rate $\eta_t \asymp \sqrt{\log(d)/t}$) enjoys $O(\sqrt{T \log d})$ worst-case regret and, as we show, has $-O(\sqrt{T \log d})$ best-case regret. Moreover, in the same setting under the restriction of two experts and binary losses, we *exactly identify the best-case sequence*, thereby showing that our best-case lower bound for this setting is tight. The structure of this sequence is surprisingly simple, but the arguments we use to pinpoint this sequence are playfully complex, bearing some similarity to the techniques of [20] and [15]. The latter work [15] considers the regret of Thompson Sampling in adversarial bit prediction; they use swapping rules and identify best-case sequences, as do we. However, the algorithms and problem settings have important differences.

A key motivation for our work is a recent result [4] which shows, in the setting of DTOL, that Hedge with constant learning rate has best-case regret lower bounded by $-O(\sqrt{T})$. This result, taken together with worst-case upper bounds of order $O(\sqrt{T})$, is then used to show that if each of finitely many experts approximately satisfies a certain notion of group fairness, then a clever use of the Hedge algorithm (running it separately on each group) also approximately satisfies the same notion of group fairness while still enjoying $O(\sqrt{T})$ regret. However, we stress that their result is very limited in that it applies only to Hedge when run with a known time-horizon. The fixed time horizon assumption also implies that their notion of group fairness also is inherently tied to a fixed time horizon (see Section 4.2 for a detailed discussion), and this latter implication can lead to experts that seem very unfair but which, based on a fixed horizon view of group fairness, are technically considered to be fair. Our best-case lower bounds enable the results of [4] to hold in much greater generality; in particular, our results enable the use of an anytime version of group fairness, which we feel is truly needed.

To our knowledge, our work is the first to study best-case lower bounds for Adaptive FTRL [16], i.e., FTRL with time-varying regularizers that can adapt to the learning algorithms' past observations. The most closely related work is a paper by Gofer and Mansour (GM) [10] which, in the setting of online linear optimization (OLO) and when using FTRL with a *fixed* regularizer,[1] provides various lower bounds on the regret. For instance, they show that for any sequence of data, the regret is nonnegative; we recover this result as a special case of our analysis, and our analysis extends to OCO as well. GM also lower bound what they call the *anytime regret*, which superficially may seem similar to our providing best-case lower bounds for anytime algorithms. Yet, as we explain in Section 3, these two notions greatly differ. In short, their analysis lower bounds the maximum regret (over all prefixes of a sequence) for fixed horizon algorithms, whereas our analysis lower bounds the regret for all prefixes (including the minimum) for adaptively regularized algorithms, which includes anytime algorithms.

A natural question is whether results similar to our results for FTRL also hold for online mirror descent (OMD). In some situations, such as in OLO when the action space is the probability simplex, the regularizer is the negative Shannon entropy, and the learning rate is constant, our results automatically apply to OMD because the methods are then the same. More generally, it is known that OMD with a time-varying learning rate can fail spectacularly by obtaining linear regret (see Theorem 4 of [18]). Since so much of our work is tied to obtaining anytime guarantees (which would require a time-varying learning rate), we forego providing best-case lower bounds for OMD.

Our main contributions are as follows:

1. We give a general best-case lower bound on the regret for Adaptive FTRL (Section 3). Our analysis crucially centers on the notion of adaptively regularized regret, which serves as a potential function to keep track of the regret.

2. We show that this general bound can easily be applied to yield concrete best-case lower bounds for FTRL with time-varying negative regularizers, one special case being the negative Shannon entropy. We also show that an adaptive gradient FTRL algorithm (which can be viewed as a "non-linearized" version of the dual averaging version of AdaGrad [8]; see Section 4.3 for details) admits a best-case lower bound that is essentially the negation of its upper bound (Section 4).

---

[1][10] uses learners that follow the gradient of a concave potential, which is essentially equivalent to FTRL.

3. A widely used variant of FTRL for OCO is to first linearize the losses, leading to linearized FTRL. This method works well with respect to upper bounds, as a basic argument involving convexity goes in the right direction. However, with regards to best-case lower bounds, we show a simple construction (Section 5) for which linearized FTRL obtains $-\Omega(T)$ regret.[2]

4. In the setting of DTOL with 2 experts and binary losses, we explicitly identify the best-case sequence, proving that our best-case lower bounds are tight in this setting (Section 6).

The next section formalizes the problem setting and FTRL. We then develop the main results.

## 2 Problem Setting and General Prediction Strategies

Before giving the problem setting, we first set some notation. We denote the norm of a vector $w \in \mathcal{W}$ as $\|w\|$, the corresponding dual norm is denoted as $\|\cdot\|_*$, and $\log$ is always the natural logarithm.

**Problem setting.** We consider the OCO setting. This is a game between Learner and Nature. In each round $t = 1, 2, \ldots, T$, Learner selects an action $w_t$ belonging to a closed, bounded, convex set $\mathcal{W} \subseteq \mathbb{R}^d$. Then, with knowledge of $w_1, \ldots, w_t$, Nature responds with a convex loss function $f_t : \mathcal{W} \mapsto \mathbb{R}$. Learner then observes $f_t$ and suffers loss $f_t(w_t)$. Learner's goal is to minimize its regret, defined as

$$\mathcal{R}_T := \sup_{w \in \mathcal{W}} \sum_{t=1}^{T} [f_t(w_t) - f_t(w)],$$

which is the gap between Learner's cumulative loss and that of the best action in hindsight.

This paper will cover several examples of OCO. The first example is the subclass of OLO problems. In OLO, the loss functions $f_t$ are linear, with $f_t(w) = \langle \ell_t, w \rangle$ for some loss vector $\ell_t \in \mathbb{R}^d$. A noteworthy special case of OLO is DTOL, also known as the Hedge setting. In DTOL, we take $\mathcal{W}$ to be equal to the simplex $\Delta_d$ over $d$ outcomes and restrict the loss vectors as $\ell_t \in [0,1]^d$. We introduce some notation that will be useful in the DTOL setting. For any expert $j \in [d]$, let $L_{t,j} := \sum_{s=1}^{t} \ell_{s,j}$ denote the cumulative loss of expert $j$ until the end of round $t$. We denote the loss of Learner in round $t$ as $\hat{\ell}_t := \langle \ell_t, w_t \rangle$ and Learner's cumulative loss at the end of round $t$ as $\hat{L}_t := \sum_{s=1}^{t} \hat{\ell}_s$.

In this work, we consider the general prediction strategy of FTRL.

**FTRL.** Let $\Phi_1, \Phi_2, \ldots$ be a possibly data-dependent sequence of regularizers where, for each $t$, the regularizer $\Phi_t$ is a mapping $\Phi_t : \mathcal{W} \to \mathbb{R}$ which is allowed to depend on $(f_s)_{s \leq t}$. Then FTRL chooses actions according to the past regularized cumulative loss:

$$w_{t+1} = \operatorname*{arg\,min}_{w \in \mathcal{W}} \left\{ \sum_{s=1}^{t} f_s(w) + \Phi_t(w) \right\}. \tag{1}$$

We would like to emphasize that this is a very general template. It includes a fixed learning rate regularization, $\Phi_t \equiv \frac{1}{\eta}\Phi$, as well as its variable learning rate counterpart, $\Phi_t = \frac{1}{\eta_t}\Phi$, and arbitrary forms of adaptive choices of $\Phi_t$ based on the past. Despite this adaptivity, in the next section we will show that proving best-case lower bounds for this strategy is quite straightforward.

The regret attained by FTRL is summarized in the following known result.

**Theorem 1** (Theorem 1 of [16]). *Let $(\Phi_t)_{t \geq 1}$ be a sequence of nonnegative regularizers such that, for each $t \geq 1$, $\Phi_t$ is 1-strongly convex with respect to a norm $\|\cdot\|_{(t)}$. The regret of the FTRL algorithm (1) with this sequence of regularizers is upper bounded as*

$$\mathcal{R}_T \leq \Phi_T(w^*) + \tfrac{1}{2} \sum_{t=1}^{T} \|\nabla f_t(w_t)\|_{(t-1),*}^2, \tag{2}$$

*where $w^* \in \mathcal{W}$ is the best action in hindsight. If we do not require the regularizers to be nonnegative but instead assume that, for each $w \in \mathcal{W}$, the sequence $(\Phi_t(w))_{t \geq 1}$ is non-increasing, then (2) still holds if we replace $\Phi_T(w^*)$ by $\Phi_T(w^*) - \inf_{w \in \mathcal{W}} \Phi_T(w)$.*

---

[2]This negative construction, combined with the fact that the dual averaging version of AdaGrad is a linearized version of FTRL, is why we only prove best-case lower bounds for the adaptive gradient FTRL algorithm.

# 3 A general best-case lower bound

We now present our general best-case lower bound for FTRL. Key to our analysis is the concept of *adaptively regularized regret* (hereafter abbreviated as "regularized regret"), defined as

$$\mathcal{R}_t^{\Phi_t} = \sup_{w \in \mathcal{W}} \Big\{ \sum_{s=1}^{t} [f_s(w_s) - f_s(w)] - \Phi_t(w) \Big\}. \tag{3}$$

The regularized regret can easily be related to the regret as

$$\mathcal{R}_t^{\Phi_t} \leq \mathcal{R}_t - \inf_{w \in \mathcal{W}} \Phi(w). \tag{4}$$

Also, applying (1), the following re-expression of the regularized regret is immediate:

$$\mathcal{R}_t^{\Phi_t} = \sum_{s=1}^{t} [f_s(w_s) - f_s(w_{t+1})] - \Phi_t(w_{t+1}). \tag{5}$$

**Theorem 2** (Best-case lower bound on regret for adaptive FTRL). *Consider the setting of online convex optimization and the adaptive FTRL strategy* (1)*. Suppose that there exists a sequence* $(\alpha_t)_{t \in [T]}$ *such that* $\Phi_t(w_t) \leq \Phi_{t-1}(w_t) + \alpha_t$ *for all* $t \in [T]$*. Then*

$$\mathcal{R}_T \geq \inf_{w \in \mathcal{W}} \Phi_T(w) - \inf_{w \in \mathcal{W}} \Phi_0(w) - \sum_{t=1}^{T} \alpha_t.$$

*Proof.* We start by inductively bounding the adaptively regularized regret:

$$\mathcal{R}_{t+1}^{\Phi_{t+1}} - \mathcal{R}_t^{\Phi_t} = \sup_{w \in \mathcal{W}} \Big\{ \sum_{s=1}^{t+1} [f_s(w_s) - f_s(w)] - \Phi_{t+1}(w) \Big\} - \sum_{s=1}^{t+1} [f_s(w_s) - f_s(w_{t+1})] + \Phi_t(w_{t+1})$$

$$\geq -\Phi_{t+1}(w_{t+1}) + \Phi_t(w_{t+1})$$

$$\geq -\alpha_{t+1},$$

where the first equality is from (5). We conclude that $\mathcal{R}_T^{\Phi_T} \geq \mathcal{R}_0^{\Phi_0} - \sum_{t=1}^{T} \alpha_t$. Next, from (4),

$$\mathcal{R}_T \geq \mathcal{R}_T^{\Phi_T} + \inf_{w \in \mathcal{W}} \Phi_T(w) \geq \inf_{w \in \mathcal{W}} \Phi_T(w) + \mathcal{R}_0^{\Phi_0} - \sum_{t=1}^{T} \alpha_t$$

$$= \inf_{w \in \mathcal{W}} \Phi_T(w) - \inf_{w \in \mathcal{W}} \Phi_0(w) - \sum_{t=1}^{T} \alpha_t,$$

where in the last equality we used that $\mathcal{R}_0^{\Phi_0} = \sup_{w \in \mathcal{W}} \{-\Phi_0(w)\}$. □

The closest results to Theorem 2 of which we are aware are in the intriguing work of Gofer and Mansour (GM) [10], who provided lower bounds for FTRL with a fixed regularizer for OLO. Their Theorem 1 shows that the best-case regret is nonnegative. One wonders if the doubling trick can extend their results to anytime best-case lower bounds; regrettably, the doubling trick's restarts can allow the algorithm to achieve -$\Omega(T)$ regret (see Appendix A). GM also lower bound a notion they call the anytime regret (see Theorem 5 of [10]). The anytime regret for a sequence as they define it is actually the maximum regret over all prefixes of the sequence (where the sequence has a fixed time horizon); ultimately, the lower bound they obtain depends on the quadratic variation of the sequence as computed on a *fixed* time horizon. Related to this, GM's analysis is for algorithms that use the same regularizer in all rounds. They lament that it is unclear how to extend their analysis to handle time-varying learning rates[3]. Our goal is rather different, as taking the maximum regret over all prefixes says nothing about how large (in the negative direction) the regret could be for some particular prefix. Thus, our style of analysis, which provides a lower bound on the regret for all time horizons (which, in particular, provides a lower bound on the *minimum* over all prefixes), differs greatly from theirs and is what is needed. We think the different styles of our work and theirs stems from the regret being nonnegative in their paper, whereas it need not be for time-varying regularizers.

Theorem 2 possesses considerable generality, owing to its applying to FTRL with adaptive regularizers. We highlight just a few applications in the next section.

---

[3]A time-varying learning rate gives rise to perhaps the most basic form of an adaptive regularizer.

# 4 Best-case lower bounds in particular settings

Theorem 2 presented in last section, despite its simplicity, is an extremely powerful method, capable of addressing several of the settings where FTRL attains sublinear regret. Next, we proceed to enumerate some important examples of instances of FTRL, comparing existing worst-case upper bounds on the regret with our best-case lower bounds.

## 4.1 Non-increasing learning rates and timeless algorithms

We first present several examples in which the elements of the sequence $(\Phi_t)_{t\geq1}$ take the form $\Phi_t = \frac{1}{\eta_t}\Phi$ for a fixed regularizer $\Phi$ and a time-varying learning rate $\eta_t$.

**Constant learning rate.** The simplest example is that of a fixed learning rate $\eta_t \equiv \eta$, which means $\Phi_t(w) = \frac{1}{\eta}\Phi(w)$. Taking $\alpha_t = 0$ for all $t$, Theorem 2 immediately implies that the regret is always nonnegative; this implication was previously shown by Gofer and Mansour (see Thm. 1 of [10]). A simple consequence is that the Follow the Leader (FTL) strategy (i.e., $\Phi_t \equiv 0$) has nonnegative regret, which also can be inferred from the results of [10]. Although we cannot find a precise reference, we believe that it was already known that FTL always obtains nonnegative regret (even prior to [10]).

**Time-varying learning rate.** More generally, taking a time-varying learning rate, Theorem 2 gives

$$\mathcal{R}_T \geq \left(\frac{1}{\eta_T} - \frac{1}{\eta_0}\right)\inf_{w\in\mathcal{W}}\Phi(w) - \sum_{t=1}^{T}\left(\frac{1}{\eta_t} - \frac{1}{\eta_{t-1}}\right)\Phi(w_{t+1}). \tag{6}$$

A typical strategy to obtain sublinear anytime worst-case regret is to set the learning rate as $\eta_t = \eta/\sqrt{t+1}$ for some constant $\eta$ that depends on various known problem-dependent constants. Continuing from (6) with this setting further implies that

$$\mathcal{R}_T \geq \frac{1}{\eta}(\sqrt{T+1} - 1)\inf_{w\in\mathcal{W}}\Phi(w) - \frac{1}{\eta}\sup_{w\in\mathcal{W}}\Phi(w)\sum_{t=1}^{T}\frac{1}{2\sqrt{t+1}}.$$

If $\Phi$ is nonpositive — as holds when $\mathcal{W}$ is the $d$-dimensional simplex and $\Phi$ is the negative Shannon entropy — the above is further lower bounded by

$$\frac{1}{\eta}(\sqrt{T+1} - 1)\inf_{w\in\mathcal{W}}\Phi(w) - \frac{1}{\eta}\sup_{w\in\mathcal{W}}\Phi(w)\sqrt{T+1}. \tag{7}$$

A particularly interesting example is the DTOL setting. In this setting, when we take $\Phi$ to be the negative Shannon entropy $\Phi(w) = \sum_{j=1}^{d} w_j \log w_j$ and set $\eta = 2\sqrt{\log d}$ so that $\eta_t = 2\sqrt{(\log d)/(t+1)}$ recovers the standard anytime version of Hedge which has also been called Decreasing Hedge [17]. In round $t$, this algorithm plays $w_t$ such that $w_{t,j} \propto \exp(-\eta_{t-1}L_{t-1,j})$ for $j \in [d]$, and we have the following anytime best-case lower bound and worst-case upper bound on the regret:

$$-\frac{1}{2}\sqrt{T\log d} \leq \mathcal{R}_T \leq \sqrt{T\log d}.$$

The lower bound holds from (7) combined with $\sqrt{T+1} - 1 \leq \sqrt{T}$ and $-\log d \leq \Phi \leq 0$, while the upper bound is from Theorem 2 of [6]. The upper bound is minimax optimal in terms of the rate and, asymptotically (letting both $d$ and $T$ go to infinity) has a constant that is optimal up to a factor of $\sqrt{2}$. As we show in Section 6, in the case of $d = 2$ the lower bound also has the optimal rate.

**Timeless algorithms.** In the context of DTOL, it is straightforward to adapt our analysis for Decreasing Hedge to Hedge with any non-increasing learning rate. One example of interest is

$$\eta_t = -\log\left(1 - \min\left\{\tfrac{1}{4}, \sqrt{\tfrac{2\log d}{L_t^*}}\right\}\right),$$

where $L_T^* = \min_{j\in[d]} L_{T,j}$ is the cumulative loss of the best expert. This choice of adaptive learning rate yields an anytime upper bound on the regret of $O\left(\sqrt{L_T^*\log d} + \log d\right)$ in the DTOL setting (see Theorem 2.1 of [2], who actually prove this result in the more general setting of prediction with

expert advice). Such a bound is called timeless because rounds in which all experts suffer the same loss have no effect on the bound [7]. This is a natural property to have in this setting, and with the above choice of learning rate, we have the following timeless[4] best-case lower bound:

$$\mathcal{R}_T \geq \min\left\{0, -\sqrt{\frac{L_T^* \log d}{2}} + 4\log d\right\};\tag{8}$$

a brief derivation is in Appendix B. Again, notice the similarity between the upper and lower bounds.

## 4.2 Group fairness in online learning

In a pioneering work, Blum, Gunasekar, Lykouris, and Srebro (BGLS) [4] considered a notion of group fairness in DTOL. In their setup, the DTOL protocol is augmented so that, at the start of each round $t$, Nature selects and reveals to Learner a group $g_t$ belonging to a set of groups $\mathcal{G}$ prior to Learner's playing its action $w_t$. They assume that for a known, fixed time horizon $T$, each expert $j \in [d]$ has balanced mistakes across groups in the following sense: For any $g \in \mathcal{G}$, let $\mathcal{T}(g) := \{t \in [T]\colon g_t = g\}$ denote the rounds belonging to group $g$, and let $L_{\mathcal{T}(g),j} := \sum_{t \in \mathcal{T}(g)} \ell_{t,j}$ denote the cumulative loss of expert $j \in [d]$ when considering only the rounds in $\mathcal{T}(g)$; then we say that expert $j$ is fair in isolation if

$$\frac{L_{\mathcal{T}(g),j}}{|\mathcal{T}(g)|} = \frac{L_{\mathcal{T}(g'),j}}{|\mathcal{T}(g')|} \quad \text{for all } g, g' \in \mathcal{G}.$$

BGLS devised a strategy based on the multiplicative weights algorithm which simultaneously satisfies an approximate notion of group fairness while still enjoying $O(\sqrt{T \log d})$ worst-case regret. The multiplicative weights algorithm (which we hereafter refer to as Hedge as the algorithms are equivalent for constant learning rate) sets $w_t$ as $w_{t,j} \propto (1 - \tilde{\eta})^{L_{t-1,j}}$ for $j \in [d]$ for a learning rate parameter $\tilde{\eta}$. Their strategy is to run a separate copy of Hedge for each group, so that for any group $g$, the copy corresponding to group $g$ is run on the subsequence corresponding to the rounds in $\mathcal{T}(g)$. For brevity, we call this "Interleaved Hedge". In BGLS's analysis (see the proof of their Theorem 3), they first give worst-case regret upper and lower bounds for Hedge. Specifically, they show that if Hedge is run with a constant learning rate, then

$$(1 - 4\tilde{\eta}) \cdot L_T^* \leq \hat{L}_T \leq (1 + \tilde{\eta})L_T^* + (\log d)/\tilde{\eta}.\tag{9}$$

An optimal, non-anytime worst-case tuning of $\tilde{\eta}$ then yields matching-magnitude regret lower and upper bounds of $-O(\sqrt{T \log d})$ and $O(\sqrt{T \log d})$ respectively. Then on the one hand, the regret of Interleaved Hedge satisfies $\mathcal{R}_T = O(\sqrt{|\mathcal{G}|T \log d})$. In addition, by virtue of BGLS's $-O(\sqrt{T \log d})$ lower bound for Hedge when fed $T$ rounds (along with the standard upper bound), their Interleaved Hedge enjoys the following group fairness guarantee:

$$\frac{\hat{L}_{\mathcal{T}(g)}}{|\mathcal{T}(g)|} - \frac{\hat{L}_{\mathcal{T}(g')}}{|\mathcal{T}(g')|} = O\left(\sqrt{\frac{\log d}{T_0}}\right) \quad \text{for all } g, g' \in \mathcal{G} \text{ and } T_0 := \min_g |\mathcal{T}(g)|,$$

where we adopt the notation $\hat{L}_{\mathcal{T}(g)} := \sum_{t \in \mathcal{T}(g)} \hat{\ell}_t$.

By using our improved nonnegative best-case lower bound for Hedge with constant learning rate (which again, is not a new result) together with an optimal, non-anytime worst-case tuning of $\eta$, we can obtain the following improvement over (9):

$$0 \leq \hat{L}_T \leq \sqrt{T \log d}.$$

As explained in the appendix (the remaining steps are essentially due to BGLS), we can then obtain the following improved group fairness guarantee:

$$\frac{\hat{L}_{\mathcal{T}(g)}}{|\mathcal{T}(g)|} - \frac{\hat{L}_{\mathcal{T}(g')}}{|\mathcal{T}(g')|} \leq \sqrt{\frac{\log d}{T_0}} \quad \text{for all } g, g' \in \mathcal{G} \text{ and } T_0 := \min_g |\mathcal{T}(g)|.$$

The above result holds when the learning rate for each group $g$ is set as $\eta^{(g)} = \sqrt{\frac{\log d}{|\mathcal{T}(g)|}}$. Of course, running Hedge instances for each group with the correct constant learning rates means that the

---

[4]Technically, the upper and lower bounds as stated are not timeless, but it is easy to see that any round for which all experts have the same loss can be replaced by a round where all experts have zero loss, with no change in the algorithm's behavior nor its regret. Our regret bounds on the modified loss sequence then become timeless.

algorithm must know $|\mathcal{T}(g)|$ for each group $g \in \mathcal{G}$, at least within a reasonable constant factor. This is a far stronger assumption than the already strong assumption of a known time horizon.

Using our anytime best-case lower bound for Decreasing Hedge, combined with the analysis of BGLS, it is straightforward to vastly extend their results in each of the following ways:

1. Using BGLS's fixed horizon notion of fairness and a copy of Decreasing Hedge for each group's instance, we can drop the assumption that each group's cardinality $|\mathcal{T}(g)|$ is known.

2. The most interesting extension, whose possibility was the original basis of our entire work, is that we can now upgrade BGLS's notion of group fairness to its anytime sibling. This involves measuring, for every prefix of the length-$T$ game, the discrepancy between the error rates of any pair of groups. This is an arguably more natural notion of fairness, as it avoids situations where an expert purports to be fair while having all of its mistakes for one group occur in the first half of the game. Since we now have an anytime best-case lower bound for Decreasing Hedge, we have the requisite piece needed to show that Interleaved (Decreasing) Hedge satisfies the same notion of anytime group fairness. Our timeless best-case lower bounds also apply here, giving that extension as well.

We now briefly sketch each of these extensions, leaving more detailed derivations to the appendix.

**Decreasing Hedge (Anytime results).** Suppose now that Interleaving Hedge uses copies of Decreasing Hedge with time-varying learning rate $\eta_t = 2\sqrt{(\log d)/(t+1)}$. Note that in this case, the copy for group $g$ increments its internal round only each time a new round for group $g$ appears. We can then automatically apply our anytime lower bound on the regret of Decreasing Hedge to obtain

$$\frac{\hat{L}_{\mathcal{T}(g)}}{|\mathcal{T}(g)|} - \frac{\hat{L}_{\mathcal{T}(g^*)}}{|\mathcal{T}(g^*)|} \leq \sqrt{\frac{\log d}{|\mathcal{T}(g)|}} + \frac{1}{2}\sqrt{\frac{\log d}{|\mathcal{T}(g^*)|}} \quad \text{for } g^* \text{ such that } \frac{\hat{L}_{\mathcal{T}(g^*)}}{|\mathcal{T}(g^*)|} = \min_{g \in \mathcal{G}} \frac{\hat{L}_{\mathcal{T}(g)}}{|\mathcal{T}(g)|}.$$

Therefore, if in hindsight we have $T_0 = \min_{g \in \mathcal{G}} |\mathcal{T}(g)|$, then the following fairness guarantee holds:

$$\frac{\hat{L}_{\mathcal{T}(g)}}{|\mathcal{T}(g)|} - \frac{\hat{L}_{\mathcal{T}(g')}}{|\mathcal{T}(g')|} \leq \frac{3}{2}\sqrt{\frac{\log d}{T_0}} \quad \text{for all } g, g' \in \mathcal{G}.$$

Note that even though Learner need not know the time horizon $T$ nor $|\mathcal{T}(g)|$ for each $g$, the above guarantee still can only hold for a fixed time horizon $T$. This is because the definition of fairness in isolation only holds for a fixed time horizon.

**Anytime fairness.** As mentioned earlier, the fixed horizon view of fairness in isolation can be very limiting. Rather than opting for fairness in isolation to hold for a fixed time horizon, we argue that it is more natural for this definition to hold in the following anytime sense:

$$\frac{L_{\mathcal{T}(g),j}}{|\mathcal{T}(g)|} = \frac{L_{\mathcal{T}(g'),j}}{|\mathcal{T}(g')|} \quad \text{for all } g, g' \in \mathcal{G} \text{ and for all } T;$$

note that the time horizon $T$ is implicit in each $\mathcal{T}(g)$.

The attentive reader may notice that this definition can be overly restrictive on Nature (perhaps impossibly so), and therefore it is natural to allow the equality to hold only approximately (which BGLS did explore). Just as in BGLS's work, it is possible to extend our results to cases where fairness holds only approximately, and such an extension is certainly warranted in the case of anytime fairness. This extension requires only straightforward modifications to the analysis and so we do not give further details here. In the case of anytime fairness, it further makes sense for approximate fairness to be defined according to a rate. We leave this extension to a future, longer paper.

Using the above anytime version of fairness in isolation, it is straightforward to extend the previous result to the following new result. Just like above, suppose now that Interleaved Hedge uses copies of Decreasing Hedge with time-varying learning rate. Then, for all time horizons $T$,[5]

$$\frac{\hat{L}_{\mathcal{T}(g)}}{|\mathcal{T}(g)|} - \frac{\hat{L}_{\mathcal{T}(g')}}{|\mathcal{T}(g')|} \leq \frac{3}{2}\sqrt{\frac{\log d}{T_0}} \quad \text{for all } g, g' \in \mathcal{G}.$$

**Mini-conclusion.** Whereas a key message of [4] is adaptive algorithms (in the sense of low shifting regret) cannot satisfy group fairness, at least when using the interleaved approach, our best-case lower bounds do cover many other types of adaptivity. This shows that with regards to group fairness and the interleaved strategy, the group-fair adversarial online learning tent is actually quite large.

---

[5]Note that in the below, each $\mathcal{T}(g)$ (and hence $T_0$) implicitly depends on the time horizon $T$.

### 4.3 Adaptive gradient FTRL

Inspired by the *adaptive gradient* (AdaGrad) algorithm for OCO [8], we consider an adaptive gradient FTRL algorithm with quadratic regularizers, $\Phi_t(w) = \frac{1}{2\eta}\langle w, H_t w\rangle$: this corresponds to the adaptive FTRL (1) with regularizers $\Phi_t$ as above. We include its two variants. In the below, we use the notation $g_t := \nabla f_t(w_t)$ and $\delta > 0$ is a fixed number:

(a) Diagonal: $H_t = \delta I + \mathrm{Diag}(s_t)$, where $s_t = \left(\left(\sqrt{\sum_{\tau=1}^t g_{j,t}^2}\right)_{j\in[d]}\right)$,

(b) Full-matrix: $H_t = \delta I + G_t^{1/2}$ where $G_t = \sum_{\tau=1}^t g_t g_t^\top$.

We emphasize that the proposed algorithm does not exactly match AdaGrad from [8]. To resolve this inconsistency, we use the regret upper bounds from Theorem 1, combined with upper bounds on the gradient norms that appear in such upper bounds, proved in [8]. This leads to the following result.

**Theorem 3** (From [8]). *Consider the setting of OCO. Given $1 \le p \le \infty$, we denote by $D_p$ the diameter of $\mathcal{W}$ in the $\|\cdot\|_p$ norm, and $M_p$ is a bound on the $\|\cdot\|_p$ norm of the gradients for any possible loss. Then the adaptive gradient FTRL algorithm satisfies the following bounds:*

*(a) Diagonal: If $\delta = M_\infty$, then $\sum_{t=1}^T \|\nabla f_t(w_t)\|_{(t-1),*}^2 \le 2\eta \sum_{j=1}^d \sqrt{\sum_{t=1}^T g_{t,j}^2}$. In particular, setting $\eta = D_\infty$, we obtain an upper bound on the regret $\frac{D_2^2}{D_\infty} M_\infty + D_\infty \sum_{j=1}^d \sqrt{\sum_{t=1}^T g_{t,j}^2}$.*

*(b) Full matrix: If $\delta = M_2$, then $\sum_{t=1}^T \|\nabla f_t(w_t)\|_{(t-1),*}^2 \le 2\eta\, \mathrm{Tr}(G_T^{1/2})$. In particular, setting $\eta = D_2$, we obtain an upper bound on the regret $D_2[M_2/2 + \mathrm{Tr}(G_T^{1/2})]$.*

We proceed to best-case lower bounds, deferring the proof to Appendix D. The proof strategy follows similar telescopic recursions to those used in [8, 16]. Notice that these bounds closely match the respective worst-case upper bounds.

**Proposition 4.** *In the OCO setting, the adaptive gradient FTRL strategy with parameter tuning as in Theorem 3 attains best-case lower bounds on the regret of $-(D_\infty/2)\sum_{j=1}^d \sqrt{\sum_{t\in[T]} g_{t,j}^2}$ in the diagonal case and $-(D_2/2)\mathrm{Tr}(G_T^{1/2})$ in the full-matrix case.*

## 5 On best-case lower bounds for other algorithms

So far, our study of best-case lower bounds has focused on adaptive FTRL algorithms. A major drawback of FTRL is that the iteration cost of each subproblem grows linearly with the number of rounds. It is then tempting to consider more efficient algorithms attaining comparable regret upper bounds. We discuss such possibilities for two natural algorithms: linearized FTRL and OMD.

### 5.1 Linearized FTRL

In this section, we give a simple construction in which linearized FTRL obtains negative linear regret, i.e., regret which is $-\Omega(T)$. The problem is a forecasting game with binary outcomes, the action space $\mathcal{D}$ equal to the 2-simplex $[0, 1]$, and the squared loss $\ell(p, y) = \frac{1}{2}(p - y)^2$. This can be cast as an OCO problem by taking $\mathcal{W} = [0, 1]$ and, for each $t \in [T]$, setting $f_t(p) = (p - y_t)^2$ for some outcome $y_t \in \{0, 1\}$ selected by Nature.

We consider linearized FTRL using the negative Shannon entropy regularizer; this is equivalent to exponentiated gradient descent [12] and uses the update $p_t = \frac{1}{1+\exp(\eta_t G_{t-1})}$, where $G_{t-1} = \sum_{s=1}^{t-1} g_s$ and each $g_s = p_s - y_s$ is the gradient of the loss with respect to $p_s$ under outcome $y_s$. In this situation, an $O(\sqrt{T})$ anytime regret upper bound can be obtained by employing the time-varying learning rate $\eta_t = 1/\sqrt{t}$ (see Theorem 2.3 of [5] or Theorem 2 of [6]).

Let Nature's outcome sequence be the piecewise constant sequence consisting of $T/2$ zeros followed by $T/2$ ones. Therefore, the best action in hindsight is $p^* = 0.5$. We now state our negative result and briefly sketch a proof; the full proof can be found in Appendix E.

**Theorem 5.** *In the construction above, linearized FTRL obtains regret $\mathcal{R}_T = -\Omega(T)$.*

*Proof (sketch).* The core idea behind the proof is that linearized FTRL exhibits a type of switching behavior. Let $q_0$ and $q_1$ be constants satisfying $0 < q_0 < 1/2 < q_1 < 1$. Observe that in the first half of the game (considered in isolation), the optimal action is 0. Thus, if $p_t$ drops below $q_0$ in the first half of the game, then for all rounds thereafter in the first half, Learner picks up negative constant regret in each round. We show that this drop happens after only a constant number of rounds for a simple reason: the gradient must be at least $q_0$ whenever $p_t > q_0$ in the first half, and $p_t$ changes according to the exponentiated cumulative gradient. Thus, Learner picks up negative linear regret in the first half of the game. Next, consider the second half of the game. For this half (considered in isolation), the optimal action is 1. Thus, once $p_t$ surpasses $q_1$, for all rounds thereafter Learner will again pick up negative constant regret. Via an arguably coarse analysis, we show that the number of rounds in the second half before Learner's action $p_t$ surpasses $q_1$ is at most a small constant fraction of $T$. To show this, we rely on the fact that whenever $p_t < q_1$ in the second half, we have that the gradient must be at least $-q_1$ (combined with the exponential effect of the gradients on $p_t$). That is, Learner quickly switches from having $p_t$ below $q_0$ to having $p_t$ above $q_1$. Thus, for most of the second half, Learner again picks up negative linear regret. □

## 5.2 Online mirror descent

For OLO over the simplex, the online mirror descent method with entropic regularization and *constant learning rate* attains best-case lower bounds $-O(\sqrt{T \log d})$ [4]. This algorithm is known to attain upper bounds on the regret of the same order. The extension of this result for *non-increasing learning rates*,[6] albeit possible, seems of limited interest, as this algorithm is known to achieve linear regret in the worst case [18], which is attributed to the unboundedness of the Bregman divergence.

## 6 Best-case loss sequence for binary DTOL with two experts

In this section, we characterize the binary loss sequence for DTOL with two experts in which the regret for Decreasing Hedge is minimized. Recall that Decreasing Hedge chooses expert $i$ at round $t$ with probability $p_{i,t} = \frac{\exp(-\eta_t L_{i,t-1})}{\sum_{j \in \{1,2\}} \exp(-\eta_t L_{j,t-1})}$, where $\eta_t = \sqrt{1/t}$. Losses are the form $\ell_t = \binom{\ell_{1,t}}{\ell_{2,t}}$ where $\ell_{1,t}, \ell_{2,t} \in \{0,1\}$. Note that $\eta_{t+1} < \eta_t$ for all $t$.

Our approach is to introduce a set of operations on a sequence of binary loss vectors $\ell_1, \ell_2, \ldots, \ell_T$ that do not increase the regret of Decreasing Hedge. Using this set of operations, we will show that any loss sequence with $T = 2K$ rounds $\ell_1, \ell_2, \ldots, \ell_{2K}$, can be converted to a specific loss sequence which we call the canonical best-case sequence.

**Definition 6.** For any length $T = 2K \geq 16$, the canonical best-case loss sequence is defined as:

$$\text{Canonical}(2K) = \binom{0}{1}^{K-1} \binom{0}{0}^2 \binom{1}{0}^{K-1}. \tag{10}$$

We now sketch our proof that (10) is indeed the best-case sequence. The full proof is in Appendix F.

First, consider an arbitrary binary loss sequence for two experts $\ell_1, \ldots, \ell_T$. We can substitute all $(1,1)$ loss vectors with $(0,0)$ loss vectors as it can be easily shown that this operation does not change the regret. Thus, there is no need for $(1,1)$ loss vectors in the best-case sequence.

Next, we introduce the notion of a leader change and show that there is no leader change in the best-case sequence. Therefore, we only need to consider loss sequences without leader changes. Formally, we say expert $j$ is a leader at round $t$ if $L_{j,t} = \min_{i \in \{1,2\}} L_{i,t}$. Moreover, if $j$ is the *only* leader at round $t$, then we call $i_t^* = j$ the strict leader in round $t$. We say that a sequence has a leader change if there exists times $t_1$ and $t_2$ where in both times the strict leader is defined[7] and $i_{t_1}^* \neq i_{t_2}^*$. Defining $\Delta_t := L_{2,t} - L_{1,t}$, observe that if a sequence has a leader change, then the sign of $\Delta$ should change at least once. We show (Lemma 15) that any loss sequence $\ell_1, \ell_2, \ldots, \ell_T$ can be converted, without increasing the regret, to a sequence $\ell_1', \ell_2', \ldots, \ell_T'$ where $\Delta$ is always nonnegative (hence, the resulting sequence has no leader change). Removing all leader changes facilitates the next operation to successfully convert loss sequences.

Next, consider the operation of swapping two consecutive loss vectors. When we swap a loss sequence $\ell_1, \ldots, \ell_{t-1}, \ell_t, \ell_{t+1}, \ell_{t+2}, \ldots, \ell_T$ at round $(t, t+1)$, the resulting sequence becomes

---

[6]In fact, we have found a simple argument showing the regret is nonnegative, strengthening the result in [4].
[7]If in round $t$, $L_{1,t} = L_{2,t}$, then the strict leader is not defined in that round.

$\boldsymbol{\ell}_1, \ldots, \boldsymbol{\ell}_{t-1}, \boldsymbol{\ell}_{t+1}, \boldsymbol{\ell}_t, \boldsymbol{\ell}_{t+2} \ldots, \boldsymbol{\ell}_T$. It can be easily shown that this swap only changes the loss of Decreasing Hedge at rounds $t$ and $t+1$. Since we only consider loss sequences with no leader change, we can assume without loss of generality that expert 1 is the only leader. Now based on the swapping rule lemma (Lemma 16), we can always move $\binom{0}{1}$ to the earlier rounds and $\binom{1}{0}$ to the later rounds. By repeatedly applying this swapping rule, the resulting loss sequence will be of the form $\boldsymbol{\ell}_1, \ldots, \boldsymbol{\ell}_T = \binom{0}{1}^a \binom{0}{0}^c \binom{1}{0}^b$. Note that as we have shown that $\Delta$ is always nonnegative, $a \geq b$.

So far, we have shown that any loss sequence can be converted to a loss sequence of the form $\boldsymbol{\ell}_1, \ldots, \boldsymbol{\ell}_T = \binom{0}{1}^a \binom{0}{0}^c \binom{1}{0}^b$. We then show in Lemma 17 that if $c$ is not even, then the sequence $\binom{0}{1}^{a-1} \binom{0}{0}^{c+1} \binom{1}{0}^b$ has smaller regret and $a - 1 \geq b$. Therefore, we only need to consider loss sequences of the form $\binom{0}{1}^a \binom{0}{0}^c \binom{1}{0}^b$ where $a \geq b$ and $c$ is even. As we assume $T$ is even, $a + b$ is also even. Now it is shown (in Lemma 18) that among all sequence of losses of form $\boldsymbol{\ell}_1, \ldots, \boldsymbol{\ell}_T = \binom{0}{1}^a \binom{0}{0}^c \binom{1}{0}^b$, where $a + b = 2K$, the loss sequence $\binom{0}{1}^K \binom{0}{0}^c \binom{1}{0}^K$ has the least regret. Therefore, we only need to consider loss sequences of this latter form where $2K + c = T$.

Finally, among all possible $(K, c)$ for loss sequences $\binom{0}{1}^K \binom{0}{0}^c \binom{1}{0}^K$ such that $2K + c = T$, we show that the regret is minimized when $c = 2$. This form coincides with the canonical best-case sequence.

**Bounding the regret.** As shown in Appendix F.6, the regret on the canonical best-case loss sequence can be lower and upper bounded as follows:

$$-\frac{e^2}{(1-\frac{1}{e})} \cdot \sqrt{T} - \frac{1}{2} \leq \mathcal{R}(T) \leq -\frac{1}{1+e^{\sqrt{2}}} \sqrt{T} + \frac{12}{\sqrt{e}} + \frac{1}{2}.$$

As both the lower and upper bounds are $-\Theta(\sqrt{T})$, the analysis in Section 3 is tight in this case.

## 7 Discussion

In this work, we provided a systematic treatment of best-case lower bounds in online learning for adaptive FTRL algorithms, discussed the impossibility of certain natural extensions, and also provided a tighter analysis of such lower bounds in the binary prediction experts setting. As one application, we have shown that our results for adaptive FTRL enable the use a broad class of adaptive online learning algorithms that satisfy a balanced mistakes notion of group fairness. Naturally, many questions still remain open, and we hope this work motivates further research in this intriguing topic. A first question relates to algorithms that can achieve negative regret: can we characterize the conditions under which this happens? The goal would be to reveal, beyond the particular example we gave in Section 5.1, structural properties of an instance that lead to substantial outperformance of the best fixed action. Returning to that example, another question arises. We have shown an example of OCO with a strongly convex loss (the squared loss) for which linearized FTRL obtains negative linear regret, whereas our best-case lower bounds for (non-linearized) FTRL with constant learning rate imply $-O(\sqrt{T})$ regret and known upper bounds imply $O(\log T)$ regret. Thus, while in this situation, linearized FTRL pays a price with respect to regret upper bounds, it can exhibit a switching behavior that might allow it to compete with a shifting sequence of comparators. Investigating this "blessing" of linearization would be a fascinating direction for future work. Finally, for DTOL with two experts, we showed that our best-case lower bounds are tight by constructing the best-case sequence. It would be interesting to show tightness without an explicit construction, as we could then say that the best-case lower bounds are achievable more generally. For instance, while our swapping-based technique is most related to that of [20] for worst-case sequences and [15] (who give both worst- and best-case sequences), it could also be worthwhile to draw from previous works [19, 13, 3, 14] which explicitly work out the minimax regret and minimax strategies. Even so, our results on identifying the structure of the best-case loss sequence show that Decreasing Hedge has the lowest regret for loss sequences for which the best shifting sequence of experts has precisely one shift. One research direction would be to see if this pattern holds more generally, for other FTRL-type algorithms.

We conclude our discussion pointing out that the theoretical nature of our work does not entail direct negative societal consequences. As far as our discussions on group fairness are concerned, one should be careful when applying these methods as the notion of balanced mistakes may be susceptible to strong forms of discrimination (see, e.g., the Discussion section in [4]). In particular, balanced mistakes is a sound notion of non-discrimination when all examples that contribute towards the performance also contribute towards fairness.

## Acknowledgments and Disclosure of Funding

CG's work was partially supported by FONDECYT Regular project 1210362, INRIA through the INRIA Associate Teams project, and ANID – Millennium Science Initiative Program – NCN17 059. NM and AM were supported by the NSERC Discovery Grant RGPIN-2018-03942. We also thank Sajjad Azami for his involvement in the earlier stages of this work.

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
