## A  Inadequacy of the doubling trick for best-case lower bounds

In this appendix, we briefly sketch in the setting of DTOL why a fixed horizon best-case lower bound (based on a constant regularizer) cannot be combined with the doubling trick in order to obtain an anytime best-case lower bound of the same order. Recall that in the standard analysis used for the doubling trick, the algorithm actually competes with the best expert over each epoch (where epoch sizes double). Consequently, it is possible that, over each epoch, an algorithm has (say) $-O(\sqrt{T})$ regret, but as compared to the single best expert over the entire sequence of rounds, the regret is $-\Omega(T)$. This can happen as follows. Let the best expert in all but the last epoch be the same expert $j_1$, while the best expert over the last epoch is another expert $j_2$. Finally, assume that (considering the cumulative loss over the full game) expert $j_1$ is linearly better than expert $j_2$, whereas in the last epoch expert $j_2$ is linearly better than expert $j_1$. In this situation, the doubling trick allows for sublinear regret with respect to the 1-shifting regret, and the best 1-shifting sequence of experts (a sequence of experts that can change experts only once) is linearly better than the best 0-shifting sequence of experts (i.e., the best constant sequence, also known as the best expert).

## B  Derivations for (8)

Recall that we set $\eta_t$ as

$$\eta_t = -\log\left(1 - \min\left\{\frac{1}{4}, \sqrt{\frac{2\log d}{L_t^*}}\right\}\right).$$

Starting from (6) and using the fact that $(\eta_t)_{t\geq 0}$ is non-increasing and $\Phi$ (the negative Shannon entropy) is non-positive, we have

$$\mathcal{R}_T \geq \left(\frac{1}{\eta_T} - \frac{1}{\eta_0}\right) \inf_{w\in\mathcal{W}} \Phi(w).$$

Next, since $-\log(1-x) \geq x$ for $x < 1$, we have that $\frac{1}{\eta_T} \leq \max\left\{4, \sqrt{\frac{L_T^*}{2\log d}}\right\}$. We also have $\eta_0 = 4$ since $L_0^* = 0$. Using these two facts, together with $\inf_{w\in\Delta_d} \Phi(w) = -\log d$ yields

$$\mathcal{R}_T \geq \left(4 - \max\left\{4, \sqrt{\frac{L_T^*}{2\log d}}\right\}\right)\log d.$$

The result (8) follows from some basics manipulations.

## C  Fairness

In this appendix, for the convenience of the reader we repeat the results from Section 4.2 while giving derivations for them along the way.

In their work, Blum, Gunasekar, Lykouris, and Srebro (BGLS) [4] express regret bounds in a somewhat different language than we do. Rather than dealing directly with the regret, they instead analyze the *approximate regret*. The $\varepsilon$-approximate regret relative to expert $j \in [d]$ is defined as

$$\hat{L}_T - (1+\varepsilon)L_{T,j}. \tag{11}$$

In order to more easily compare to their results, and to interpret their results in the large body of literature that focuses on the actual (non-approximate) regret, we first show how to modify some of their analysis to give results based on the actual regret (hereafter simply referred to as the "regret", just like in the main text).

First, recall that BGLS use the multiplicative weights algorithm, which sets $w_t$ as $w_{t,j} \propto (1-\tilde{\eta})^{L_{t-1,j}}$ for $j \in [d]$ for a learning rate parameter $\tilde{\eta}$. In BGLS's proof of their Theorem 3, they first give worst-case regret upper and lower bounds for Hedge ("multiplicative weights" in their work). Specifically, they show that

$$(1 - 4\tilde{\eta}) \cdot L_T^* \leq \hat{L}_T \leq (1+\tilde{\eta})L_T^* + \frac{\log d}{\tilde{\eta}}.$$

An optimal, non-anytime worst-case tuning of $\tilde{\eta}$ then yields matching-magnitude regret lower and upper bounds of $-O(\sqrt{T \log d})$ and $O(\sqrt{T \log d})$ respectively. We note in passing that the anytime version of multiplicative weights (which uses incremental updates) is equivalent to OMD with a time-varying learning rate and is known to achieve linear regret ([18, Theorem 4]).

We can obtain a similar bound for constant learning rate Hedge with an optimal, non-anytime worst case tuning of $\eta$ (defined as in the main text of our paper), with the improvement[8]

$$0 \leq \hat{L}_T \leq \sqrt{T \log d}.$$

Next, we briefly explain how to modify BGLS's proof of their Theorem 3. We first need some notation (largely derived from BGLS, but with small modifications to integrate more nicely into our notation). For any group $g$, let $j^*(g)$ be the best expert when considering the rounds involving group $g$ (i.e., $\mathcal{T}(g)$). Therefore, $j^*(g) \in \arg\min_{j \in [d]} L_{\mathcal{T}(g),j}$. Next, recall that Interleaving Hedge runs a separate copy of Hedge for each group. Let $g^*$ be the group whose copy of Hedge obtains the lowest average loss. That is, $g^*$ is such that

$$\frac{\hat{L}_{\mathcal{T}(g^*)}}{|\mathcal{T}(g^*)|} = \min_{g \in \mathcal{G}} \frac{\hat{L}_{\mathcal{T}(g)}}{|\mathcal{T}(g)|}.$$

We now pick up at the last math display in Appendix C of BGLS [4]; this is the step where the lower and upper regret bounds are used. Adjusting their analysis using our bounds, we have for a fixed time horizon $T$ and when the copy of Hedge running on group $g$ uses learning rate[9] $\eta^{(g)} = \sqrt{\frac{\log d}{|\mathcal{T}(g)|}}$:

$$\frac{\hat{L}_{\mathcal{T}(g)}}{|\mathcal{T}(g)|} - \frac{\hat{L}_{\mathcal{T}(g^*)}}{|\mathcal{T}(g^*)|} \leq \frac{L_{\mathcal{T}(g),j^*(g)}}{|\mathcal{T}(g)|} + \sqrt{\frac{\log d}{|\mathcal{T}(g)|}} - \frac{L_{\mathcal{T}(g^*),j^*(g^*)}}{|\mathcal{T}(g^*)|}$$

$$\leq \frac{L_{\mathcal{T}(g),j^*(g^*)}}{|\mathcal{T}(g)|} - \frac{L_{\mathcal{T}(g^*),j^*(g^*)}}{|\mathcal{T}(g^*)|} + \sqrt{\frac{\log d}{|\mathcal{T}(g)|}}$$

$$= \sqrt{\frac{\log d}{|\mathcal{T}(g)|}},$$

where the first inequality uses the regret lower and upper bounds, the second inequality is based on the optimality of $j^*(g)$ for group $g$, and the equality uses fairness in isolation (for any pair of groups $g, g' \in \mathcal{G}$ and expert $j$ (including $j^*(g^*)$), we have that $\frac{L_{\mathcal{T}(g),j}}{|\mathcal{T}(g)|} = \frac{L_{\mathcal{T}(g'),j}}{|\mathcal{T}(g')|}$.

**Decreasing Hedge (Anytime analysis).** Suppose now that Interleaving Hedge uses copies of Decreasing Hedge with time-varying learning rate $\eta_t = 2\sqrt{(\log d)/(t+1)}$. Note that in this case, the copy for group $g$ increments its internal round only each time a new round for group $g$ appears. We can then automatically apply our anytime lower bound on the regret of Decreasing Hedge (together with the already well-known regret upper bound) to obtain

$$\frac{\hat{L}_{\mathcal{T}(g)}}{|\mathcal{T}(g)|} - \frac{\hat{L}_{\mathcal{T}(g^*)}}{|\mathcal{T}(g^*)|} \leq \frac{L_{\mathcal{T}(g),j^*(g)}}{|\mathcal{T}(g)|} + \sqrt{\frac{\log d}{|\mathcal{T}(g)|}} - \frac{L_{\mathcal{T}(g^*),j^*(g^*)}}{|\mathcal{T}(g^*)|} + \frac{1}{2}\sqrt{\frac{\log d}{|\mathcal{T}(g^*)|}}$$

$$\leq \frac{L_{\mathcal{T}(g),j^*(g^*)}}{|\mathcal{T}(g)|} - \frac{L_{\mathcal{T}(g^*),j^*(g^*)}}{|\mathcal{T}(g^*)|} + \sqrt{\frac{\log d}{|\mathcal{T}(g)|}} + \frac{1}{2}\sqrt{\frac{\log d}{|\mathcal{T}(g^*)|}}$$

$$= \sqrt{\frac{\log d}{|\mathcal{T}(g)|}} + \frac{1}{2}\sqrt{\frac{\log d}{|\mathcal{T}(g^*)|}}.$$

Therefore, if in hindsight we have $T_0 := \min_{g \in \mathcal{G}} |\mathcal{T}(g)|$, then the following fairness guarantee holds:

$$\frac{\hat{L}_{\mathcal{T}(g)}}{|\mathcal{T}(g)|} - \frac{\hat{L}_{\mathcal{T}(g^*)}}{|\mathcal{T}(g^*)|} \leq \frac{3}{2}\sqrt{\frac{\log d}{T_0}}.$$

---

[8]Since this is a fixed horizon setting for now, the constant in the upper bound can be improved, but for simplicity we will just use the constant of 1.

[9]Yes, it is a very strong assumption to assume that the $\mathcal{T}(g)$'s are known ahead of time, but this is precisely our point. We will relax this soon.

Note that even though Learner need not know the time horizon $T$ nor $|\mathcal{T}(g)|$ for each $g$, the above guarantee still can only hold for a fixed time horizon $T$. This is because the definition of fairness in isolation only holds for a fixed time horizon. By adjusting this definition as we did in the main text (which we believe is sensible), we can extend this result to hold for anytime fairness.

## D   Proof of Proposition 4

Before proceeding to the proof of this result, notice that without loss of generality, we may assume that $0 \in \mathcal{W}$ (if not, shift the regularizers by centering them at any fixed point $\bar{w} \in \mathcal{W}$). This implies that $\inf_{w \in \mathcal{W}} \Phi_t(w) = 0$, so we only need to focus on the coefficients $\alpha_t$.

*Proof.*     1. **Diagonal case.** First, for the diagonal case, let $s_t$ be the $d$-dimensional vector with coefficients $s_{t,j} = \sqrt{\sum_{s=1}^{t} g_{s,j}^2}$. Then, if we let $\mathbf{1} \in \mathbb{R}^d$ be the all-ones vector,

$$
\begin{aligned}
\alpha_{t+1} = \langle w_{t+1}, (H_{t+1} - H_t)w_{t+1}\rangle &= \langle w_{t+1}, \mathrm{Diag}(s_{t+1} - s_t)w_{t+1}\rangle \\
&\le \max_{j \in [d]} (w_j^{t+1})^2 \|s_{t+1} - s_t\|_1 = \max_{j \in [d]} (w_j^{t+1})^2 \langle s_{t+1} - s_t, \mathbf{1}\rangle,
\end{aligned}
$$

where we used that $s_t$ is coordinate-wise nondecreasing. Next,

$$
\sum_{t=1}^{T} \alpha_t = \frac{1}{2\eta} \sum_{t=0}^{T-1} \langle w_{t+1}, (H_{t+1} - H_t)w_{t+1}\rangle \le \frac{1}{2\eta} \max_{t \in [T]} \|w_t\|_\infty^2 \langle s_T - s_0, \mathbf{1}\rangle.
$$

Hence, choosing $\eta = D_\infty$, we obtain a lower bound on regret $-\frac{1}{2}D_\infty \sum_{j=1}^{d} \sqrt{\sum_{t=1}^{T} g_{t,j}^2}$.

2. **Full-matrix case.** Now for the full-matrix update, we can proceed similarly. First,

$$
\begin{aligned}
2\eta\alpha_{t+1} = \langle w_{t+1}, (H_{t+1} - H_t)w_{t+1}\rangle &= \langle w_{t+1}, (G_{t+1}^{1/2} - G_t^{1/2})w_{t+1}\rangle \\
&\le \|w_{t+1}\|_2^2\, \lambda_{\max}(G_{t+1}^{1/2} - G_t^{1/2}) = \|w_{t+1}\|_2^2\, \mathrm{Tr}(G_{t+1}^{1/2} - G_t^{1/2}),
\end{aligned}
$$

where we used that the matrix $G_{t+1}^{1/2} - G_t^{1/2}$ is positive semidefinite. Now, summing over $t$,

$$
\sum_{t=1}^{T} \alpha_t = \frac{1}{2\eta} \sum_{t=1}^{T} \langle w_{t+1}, (H_{t+1} - H_t)w_{t+1}\rangle \le \frac{1}{2\eta} \max_{t \in [T]} \|w_t\|_2^2 \left( \mathrm{Tr}(G_T^{1/2}) - \mathrm{Tr}(G_0^{1/2}) \right).
$$

To conclude, for $\eta = D_2$, the regret is lower bounded by

$$
\mathcal{R}^T \ge -\frac{D_2}{2} \mathrm{Tr}(G_T^{1/2}),
$$

which proves the result.

$\qquad \square$

## E   Negative result for Linearized FTRL

In this section, we give a full proof of Theorem 5.

*Proof (of Theorem 5).* Let $q_0$ and $q_1$ be constants satisfying $0 < q_0 < 1/2 < q_1 < 1$; we will tune these constants later. For the analysis, we divide the $T$ rounds into 4 segments: *(i)* the rounds in the first half for which $p_t \ge q_0$; *(ii)* the remaining rounds in the first half; *(iii)* the rounds in the second half for which $p_t \le q_1$; *(iv)* the remaining rounds in the second half.

The basic idea of the proof is to show that $p_t < q_0$ after a constant number of rounds (where the constant depends on $q_0$) and hence the first segment is of constant length. In the second segment, the algorithm picks up negative linear regret since $p_t < q_0 < 1/2 = p^*$ and $y_t = 0$ in the second segment. In the third segment, where the outcomes now satisfy $y_t = 1$, we show that the algorithm can take at most a linear number of rounds (but with a suitably small coefficient) before satisfying

$p_t > q_1$ and so picks up at most linear positive regret (but again, with a suitably small coefficient). Finally, in the last segment, the algorithm once again picks up negative linear regret since in this segment $y_t = 1$ and $p_t > q_1 > 1/2 = p^*$ and also, as we show, this segment is linear in length.

In more detail, we will upper bound the regret via the following claims, whose proofs appear after this proof.

**Claim 1:** In the first half of the game, we always have $p_t \leq \frac{1}{2}$.

A simple consequence of Claim 1 is that the first segment's contribution to the regret is nonpositive (recall that $y_t = 0$ in the first half).

**Claim 2:** The number of rounds in the first segment is at most $t_1 = O(1)$.

**Claim 3:** The number of rounds in the third segment is at most $t_3 = \frac{q_0}{2(1-q_1)}T + O(\sqrt{T})$.

Putting the 3 claims together and recalling that the contribution to the regret is nonpositive in the first segment, we have that the regret of the algorithm is at most:

$$\underbrace{-\left(\frac{T}{2} - t_1\right) \cdot \left(\frac{1}{8} - \frac{q_0^2}{2}\right)}_{\text{second segment}} + \underbrace{t_3 \cdot \left(\frac{1}{2} - \frac{1}{8}\right)}_{\text{third segment}} - \underbrace{\left(\frac{T}{2} - t_3\right) \cdot \left(\frac{1}{8} - \frac{(1-q_1)^2}{2}\right)}_{\text{fourth segment}}. \qquad (12)$$

Substituting the values of $t_1$ and $t_3$, grouping terms, and rearranging, the above is equal to

$$\frac{T}{4}\left(-\frac{1}{2} + q_0^2 + \frac{q_0}{1-q_1} + (1-q_1)^2 - q_0(1-q_1)\right) + O(\sqrt{T}).$$

A suitable choice of $q_0$ is $q_0 = (1-q_1)^2$, which yields

$$\frac{T}{4}\left(-\frac{1}{2} + (1-q_1)^4 + (1-q_1) + (1-q_1)^2 - (1-q_1)^3\right) + O(\sqrt{T})$$

$$\leq \frac{T}{4}\left(-\frac{1}{2} + (1-q_1) + (1-q_1)^2\right) + O(\sqrt{T}).$$

We can take the moderate choice $q_1 = \frac{3}{4}$ for example (and hence $q_0 = \frac{1}{16}$), yielding the upper bound $-\frac{3T}{64} + O(\sqrt{T})$. $\qquad \square$

We now prove the claims.

*Proof (of Claim 1).* Consider an arbitrary round $t$ in the first half. Observe that $g_s = p_s \geq 0$ for all $s < t$; consequently, $G_{t-1} \geq 0$. The claim follows from the definition of $p_t$. $\qquad \square$

*Proof (of Claim 2).* We will show that the number of rounds in the first segment is at most

$$t_1 := \left\lceil \left(\frac{1}{2q_0}\left(\log\frac{1-q_0}{q_0} + \sqrt{\log^2\frac{1-q_0}{q_0} + 4q_0^2}\right)\right)^2 \right\rceil.$$

The idea behind the proof is that whenever $p_t \geq q_0$, the gradient $g_t$ will be sufficiently positive and, consequently, when $p_t$ is updated it will make sufficient progress towards dropping below $q_0$. We now formalize this intuition.

Let $\mathcal{T}_1 \subseteq [T/2]$ be the set of round indices in the first segment. By definition of the first segment, we have $p_t \geq q_0$ for all $t \in \mathcal{T}_1$. Therefore, for $t$ such that $t - 1 \in \mathcal{T}_1$,

$$p_t = \frac{1}{1 + \exp\left(\eta_t \sum_{s=1}^{t-1} g_t\right)}$$

$$\leq \frac{1}{1 + \exp\left(\eta_t \sum_{s=1}^{t-1} q_0\right)} = \frac{1}{1 + \exp\left(q_0 \frac{t-1}{\sqrt{t}}\right)}. \qquad (13)$$

It suffices to find the smallest $t$ such that (13) is at most $q_0$. This is equivalent to finding the smallest $t$ such that

$$q_0 \frac{t-1}{\sqrt{t}} \geq \log \frac{1-q_0}{q_0}.$$

Rearranging and making the replacement $u := \sqrt{t}$, it suffices to solve the quadratic equation

$$q_0 u^2 - \left(\log \frac{1-q_0}{q_0}\right) u - q_0 = 0.$$

Taking the positive solution and rounding up yields that the smallest such $t$ is at most

$$\left\lceil \left(\frac{\log \frac{1-q_0}{q_0} + \sqrt{\log^2 \frac{1-q_0}{q_0} + 4q_0^2}}{2q_0}\right)^2 \right\rceil = t_1.$$

$\square$

*Proof (of Claim 3).* Let $\mathcal{T}_3$ be the set of round indices in the third segment. Let $t$ be such that $t - 1 \in \mathcal{T}_3$ (note: this implies that $t - 1 \geq T/2$). Then Claims 1 and 2, together with the definition of segment 3, imply that

$$p_t = \frac{1}{1 + \exp\left(\eta_t \sum_{s=1}^{t-1} g_t\right)}$$

$$\geq \frac{1}{1 + \exp\left(\eta_t \left(t_1 \cdot \frac{1}{2} + \left(\frac{T}{2} - t_1\right) \cdot q_0 - \left(t - 1 - \frac{T}{2}\right) \cdot (1 - q_1)\right)\right)}. \tag{14}$$

It suffices to find the smallest $t$ such that (14) is at least $q_1$. Proceeding similarly to the proof of Claim 2 (and once again introducing $u := \sqrt{t}$ yields the quadratic equation

$$-(1 - q_1)u^2 + \left(\log \frac{q_1}{1 - q_1}\right) u + \left(\frac{1 - q_1 + q_0}{2} \cdot T + t_1 \left(\frac{1}{2} - q_0\right) + 1 - q_1\right) = 0$$

Solving for $u$ yields

$$u = \frac{\log \frac{q_1}{1-q_1} + \sqrt{\log^2 \frac{q_1}{1-q_1} + 4(1 - q_1)\left(\frac{1 - q_1 + q_0}{2} \cdot T + t_1 \left(\frac{1}{2} - q_0\right) + 1 - q_1\right)}}{2(1 - q_1)}.$$

From the above, we see that the smallest $t$ satisfying (14) is equal to

$$\frac{1 - q_1 + q_0}{2(1 - q_1)} \cdot T + O\left(\sqrt{T}\right),$$

which proves the claim. $\square$

## F  Best-case loss sequence for binary DTOL with 2 experts

In this section, we characterize the binary loss sequence for DTOL with two experts in which the regret for Decreasing Hedge is minimized. Recall that Decreasing Hedge chooses expert $i$ at round $t$ with probability $p_{i,t} = \frac{\exp(-\eta_t L_{i,t-1})}{\sum_{j \in \{1,2\}} \exp(-\eta_t L_{j,t-1})}$, where $\eta_t = \sqrt{1/t}$. We denote the loss vector for round $t$ as $\ell_t = \binom{\ell_{1,t}}{\ell_{2,t}}$, where $\ell_{1,t}, \ell_{2,t} \in \{0, 1\}$. Note that $\eta_{t+1} < \eta_t$ for all $t$. We denote $\binom{L_{t,1}}{L_{t,2}}$ by $L_t$.

Our approach is to introduce a set of operations on a sequence of binary losses $\ell_1, \ell_2, \ldots, \ell_T$ that do not increase the regret of Decreasing Hedge. Using this set of operations, we will show that any (suitably long) loss sequence with $T = 2K$ rounds $\ell_1, \ell_2, \ldots, \ell_{2K}$, can be converted to a specific loss sequence which we call the canonical best-case sequence.

**Definition 7.** For any length $2K \geq 16$, the canonical best case loss sequence will be defined as follows.

$$\text{Canonical}(2K) = \binom{0}{1}^{K-1} \binom{0}{0}^2 \binom{1}{0}^{K-1}. \tag{15}$$

In this appendix, we will give a complete presentation of Section 6 along with complete proofs.

### F.1 Converting all (1,1) to (0,0)

In this part, we will show that adding a constant term to a loss sequence at round $t$ would not have any effect on the regret. This is because in Decreasing Hedge, the weights in each round $t$ depend only on the difference between the cumulative loss of two experts up until the end of round $t - 1$. Adding a constant loss to both experts at any round would not change the difference between cumulative losses at any round; therefore, the weight vector remains the same for all rounds. Moreover, in round $t$, the best expert as well as Decreasing Hedge incur the same additional loss; therefore, the regret would be the same in that round.

**Lemma 8.** *Adding constant value $c$ to the loss of all experts at round $t$ would not have any effect on the regret.*

Using Lemma 8, we can substitute all (1,1) loss vectors with (0,0) loss vectors. This means that there is no need for (1,1) loss vectors in the best-case sequence.

### F.2 Dealing with Leader Change

We define $\bar{i} := \begin{cases} 2 & i = 1 \\ 1 & i = 2 \end{cases}$ and let $L_{i,t}$ denote the cumulative loss of expert $i$ up until the end of round $t$. We say expert $j$ is a leader at round $t$ if $L_{j,t} = \min_{i \in \{1,2\}} L_{i,t}$. Moreover, if $j$ is the *only* leader at round $t$, then we define the *strict leader* at round $t$ to be $i_t^* = j$.[10]

We say that a sequence has a leader change if there exists times $t_1$ and $t_2$ such that in both times the strict leader is defined and $i_{t_1}^* \neq i_{t_2}^*$. Defining $\Delta_t := L_{2,t} - L_{1,t}$, observe that if a sequence has a leader change, then the sign of $\Delta$ should change at least once. In the following, we will show that any loss sequence $\ell_1, \ell_2, \ldots, \ell_T$ can be converted, without increasing the regret, to a sequence $\ell'_1, \ell'_2, \ldots, \ell'_T$ where $\Delta$ is always nonnegative; therefore the resulting sequence does not have any leader change.

We first need to define an operation called switching the loss at round $t$.

**Definition 9.** The operation of *switching loss sequence* $\ell_1, \ell_2, \ldots, \ell_T$ *at round* $t$ yields a new loss sequence $\ell'_1, \ell'_2, \ldots, \ell'_T$ where for all $s < t$, we have $\ell'_{i,s} = \ell_{i,s}$ and for all $s \geq t$, we have $\ell'_{i,s} = \ell_{\bar{i},s}$.

Obviously switching loss sequence $\ell_1, \ell_2, \ldots, \ell_T$ at round 1 only swaps the indices of the experts; therefore, the regret remains the same for the switched loss sequence. Similarly, we will show that for any $t$, where $\Delta_{t-1} = 0$, switching loss sequence $\ell_1, \ell_2, \ldots, \ell_T$ at round $t$ does not change the regret.

**Lemma 10.** *For a loss sequence $\ell_1, \ell_2, \ldots, \ell_T$, if $\Delta_{t-1} = 0$ then switching loss sequence at round $t$ does not change the regret.*

*Proof.* We show that after switching at round $t$ where $\Delta_{t-1} = L_{2,t-1} - L_{1,t-1} = 0$, the cumulative loss of Decreasing Hedge and the best expert remains the same. Therefore, the regret does not change.

For any round $k \geq t$,

$$L'_{i,k} := \sum_{s=1}^{t-1} \ell'_{i,s} + \sum_{s=t}^{k} \ell'_{i,s} = \sum_{s=1}^{t-1} \ell_{i,s} + \sum_{s=t}^{k} \ell_{\bar{i},s} \qquad \text{(switched loss definition)}$$

$$= L_{i,t-1} + \sum_{s=t}^{k} \ell_{\bar{i},s} = L_{\bar{i},t-1} + \sum_{s=t}^{k} \ell_{\bar{i},s} \qquad (\Delta_{t-1} = 0)$$

$$= L_{\bar{i},k}.$$

Hence, for $k = T$ we have $L'_{1,T} = L_{2,T}$, and $L'_{2,T} = L_{1,T}$. Therefore, the cumulative loss of the best expert for both loss sequences is the same, but the index of the best expert is changed. Moreover, observe that for any $k \geq t$,

$$\Delta'_k := L'_{2,k} - L'_{1,k} = L_{1,k} - L_{2,k} = -\Delta_k.$$

---

[10]If in round $t$, $L_{1,t} = L_{2,t}$, then the strict leader is not defined in that round.

Also for round $t$ we know that
$$\Delta'_{t-1} := L'_{2,t-1} - L'_{1,t-1} = L_{2,t-1} - L_{1,t-1} = \Delta_{t-1} = 0 = -\Delta_{t-1}.$$

Therefore, for $s$ such that $s \geq t$, we have $\Delta'_{s-1} = -\Delta_{s-1}$.

Next, denote the loss of Decreasing Hedge for loss sequence $\boldsymbol{\ell}_1, \ldots, \boldsymbol{\ell}_T$ (and $\boldsymbol{\ell}'_1, \ldots, \boldsymbol{\ell}'_T$) at round $s$ by $\hat{\ell}_s$ (and $\hat{\ell}'_s$ respectively). Obviously, as there is no difference up until round $t$, for all rounds $s < t$, we have $\hat{\ell}_s = \hat{\ell}'_s$. Now, for any round $s \geq t$:

$$\begin{aligned}
\hat{\ell}'_s &= \sum_{i \in \{1,2\}} \frac{\exp(-\eta_s(L'_{i,s-1}))}{\exp(-\eta_s(L'_{i,s-1})) + \exp(-\eta_s(L'_{\bar{i},s-1}))} \ell'_{i,s} \\
&= \sum_{i \in \{1,2\}} \frac{1}{1 + \exp\left(\eta_s(L'_{i,s-1} - L'_{\bar{i},s-1})\right)} \ell'_{i,s} \\
&= \frac{1}{1 + \exp\left(\eta_s(L'_{1,s-1} - L'_{2,s-1})\right)} \ell'_{1,s} + \frac{1}{1 + \exp\left(\eta_s(L'_{2,s-1} - L'_{1,s-1})\right)} \ell'_{2,s} \\
&= \frac{1}{1 + \exp\left(\eta_s(-\Delta'_{s-1})\right)} \ell'_{1,s} + \frac{1}{1 + \exp\left(\eta_s(\Delta'_{s-1})\right)} \ell'_{2,s} \\
&= \frac{1}{1 + \exp\left(\eta_s(\Delta_{s-1})\right)} \ell_{2,s} + \frac{1}{1 + \exp\left(\eta_s(-\Delta_{s-1})\right)} \ell_{1,s} \\
&= \sum_{i \in \{1,2\}} \frac{1}{1 + \exp\left(\eta_s(L_{i,s-1} - L_{\bar{i},s-1})\right)} \ell_{i,s} = \hat{\ell}_s.
\end{aligned}$$

Therefore, the cumulative loss of Decreasing Hedge for both loss sequences is the same. $\qquad \square$

We also need to formally define the notion of a leader change at round $t$.

**Definition 11.** We say a loss sequence has a leader change at round $t$ if there exists $k < t$ such that for all $s$ satisfying $t - k < s < t$, it holds that $\Delta_s = 0$ and $\Delta_t \cdot \Delta_{t-k} < 0$.

**Definition 12.** Consider a sequence $\boldsymbol{\ell}_1, \boldsymbol{\ell}_2, \ldots, \boldsymbol{\ell}_T$ where $t$ is the smallest value such that $\Delta_t \neq 0$. We define the first strict leader in this sequence to be expert 1 if $\Delta_t > 0$ and expert 2 if $\Delta_t < 0$.

We are now ready to show that any sequence with leader changes can be converted, without increasing the regret, to a sequence with no leader change. We only consider loss sequences in which the first strict leader is expert 1 because if the first strict leader for a sequence $\boldsymbol{\ell}_1, \boldsymbol{\ell}_2, \ldots, \boldsymbol{\ell}_T$ is expert 2, then we can simply swap the experts' indices.

Now, if the first strict leader is expert 1, then $\Delta$ first becomes positive before it gets any chance to become negative. Our goal now is to modify the loss sequence without increasing the regret so that $\Delta$ stays nonnegative for all rounds. The number of times the sign of $\Delta$ changes corresponds to the number of leader changes.

**Observation 13.** The number of leader changes in a sequence is the number of times the sign of $\Delta_t$ changes from negative to positive or vice versa.

**Lemma 14.** *If a loss sequence $\boldsymbol{\ell}_1, \ldots, \boldsymbol{\ell}_T$ has at least one leader change, and the first leader change happens at round $t$, then switching the loss sequence at round $t$ would remove the first leader change without changing the regret. Moreover, this switch operation ensures that the first leader change (if any) in resulting loss sequence $\boldsymbol{\ell}'_1, \ldots, \boldsymbol{\ell}'_T$ happens at time $s > t$.*

*Proof.* We assume that the first strict leader is expert 1, therefore $\Delta_s \geq 0$ for $s < t$. Now, in round $t$ where the first leader change happens, we know that $\Delta_t < 0$. Moreover, $\Delta_{t-1} = 0$. Now by Lemma 10 we can switch the loss sequence at round $t$ without increasing the regret. In the resulting loss sequence, for all $s < t$, $\Delta'_s = \Delta_s$ and for all $s \geq t$, $\Delta'_s = -\Delta_s$. This ensures that for all $s \leq t$, $\Delta'_s \geq 0$. Therefore, in the resulting loss sequence, no leader change could happen at any time $s \leq t$. Therefore, the first leader change (if any) in the resulting loss sequence happens at some time $s > t$. $\qquad \square$

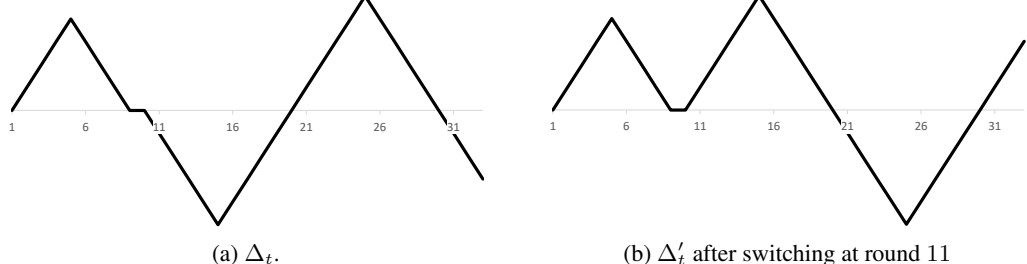

(a) $\Delta_t$.    (b) $\Delta_t'$ after switching at round 11

Figure 1: The plots show $\Delta_t$ and $\Delta_t'$ for a loss sequence $\ell_1, \ldots, \ell_T$ and $\ell_1', \ldots, \ell_T'$. The first leader change in $\ell_1, \ldots, \ell_T$ happens at round 11. Switching at round 11 decreases the number of leader changes. The first leader change in the resutling sequence $\ell_1', \ldots, \ell_T'$ happens at round $21 > 11$.

**Lemma 15.** *Any loss sequence with a leader change can be converted to a loss sequence with no leader change without changing the regret.*

*Proof.* If a sequence has at least one leader change, we can remove the first leader change at round $t$ using Lemma 14. By applying this lemma, the first leader change (if any) in the resulting loss sequence happens at round $s > t$. Because the number of rounds $T$ is bounded, we can use this operation finitely many times to reach a sequence without any leader change. $\qquad\square$

### F.3   Swapping two consecutive losses

In this section, we will consider the operation of swapping two consecutive loss vectors. When we swap a loss sequence $\ell_1, \ldots, \ell_{t-1}, \ell_t, \ell_{t+1}, \ell_{t+2}, \ldots, \ell_T$ at rounds $(t, t+1)$, the resulting sequence becomes $\ell_1, \ldots, \ell_{t-1}, \ell_{t+1}, \ell_t, \ell_{t+2} \ldots, \ell_T$. Because the behavior of Decreasing Hedge at round $s$ only depends on $\boldsymbol{L}_{s-1}$, swapping two consecutive loss vectors at rounds $(t, t+1)$ does not change $\boldsymbol{L}_{s-1}$ for $s \in [T] \setminus \{t, t+1\}$ and therefore does not affect the behavior of the algorithm at any rounds other than rounds $t$ and $t+1$. Therefore, in order to see the effect of swapping two consecutive loss vectors at rounds $(t, t+1)$, it is enough to compare $\hat{\ell}_t + \hat{\ell}_{t+1}$ with $\hat{\ell}_t' + \hat{\ell}_{t+1}'$ where $\hat{\ell}_t'$ is defined to be the loss incurred by Decreasing Hedge at round $t$ in the modified sequence.

In Section F.1, we had shown that we can replace all $\binom{1}{1}$ loss vectors with $\binom{0}{0}$ without changing the regret. Thus, we only consider all possible sequences with loss vectors $\binom{1}{0}$, $\binom{0}{1}$, $\binom{0}{0}$ and characterize when swapping two consecutive loss vectors of the above form does not increase the regret.

Moreover, by Section F.2, we only need to consider loss sequences $\ell_1, \ldots, \ell_T$ that do not have a leader change prior to the swap. Moreover, without loss of generality, we can assume the expert 1 is always the leader in the loss sequence.

We will now show that in a sequence where expert 1 is always the leader, any swap that pushes the $\binom{1}{0}$ vector to the later rounds or any swap that pushes the $\binom{0}{1}$ to the earlier rounds does not increase the regret.

**Lemma 16** (Swapping rules). *In a loss sequence $\ell_1, \ldots, \ell_T$, where $\Delta_{t-1} \geq 0$, swapping at round $(t, t+1)$ does not increase the regret in any of the following cases:*

*(a) $\ell_t, \ell_{t+1} = \binom{1}{0}\binom{0}{0}$;*

*(b) $\ell_t, \ell_{t+1} = \binom{0}{0}\binom{0}{1}$;*

*(c) $\ell_t, \ell_{t+1} = \binom{1}{0}\binom{0}{1}$.*

*Proof.* Let $p_t$ (and $p_t'$) be the probability of choosing arm 1 in round $t$ in loss sequence $\ell_1, \ldots, \ell_T$ (and $\ell_1', \ldots, \ell_T'$ respectively). Define an increasing function $f(x) := \frac{1}{1+\exp(-x)}$. Now, we can rewrite $p_t$ as follows:

$$p_t = \frac{1}{1 + \exp(-\eta_t \Delta_{t-1})} = f(\eta_t \Delta_{t-1}).$$

(a) In this case, the loss incurred by Decreasing Hedge for loss sequence $\ell_1, \ldots, \ell_T$ at rounds $t$ and $t+1$ can be expressed as

$$\hat{\ell}_t + \hat{\ell}_{t+1} = \underbrace{\begin{pmatrix} p_t \\ 1 - p_t \end{pmatrix} \cdot \begin{pmatrix} 1 \\ 0 \end{pmatrix}}_{\text{round } t} + \underbrace{\begin{pmatrix} p_{t+1} \\ 1 - p_{t+1} \end{pmatrix} \cdot \begin{pmatrix} 0 \\ 0 \end{pmatrix}}_{\text{round } t+1} = p_t.$$

Similarly the loss incurred by Decreasing Hedge for loss sequence $\ell'_1, \ldots, \ell'_T$ at rounds $t$ and $t+1$ can be written as

$$\hat{\ell}'_t + \hat{\ell}'_{t+1} = \underbrace{\begin{pmatrix} p'_t \\ 1 - p'_t \end{pmatrix} \cdot \begin{pmatrix} 0 \\ 0 \end{pmatrix}}_{\text{round } t} + \underbrace{\begin{pmatrix} p'_{t+1} \\ 1 - p'_{t+1} \end{pmatrix} \cdot \begin{pmatrix} 1 \\ 0 \end{pmatrix}}_{\text{round } t+1} = p'_{t+1}.$$

We know that Decreasing Hedge incurs the same loss at any round $s \in [T] \setminus \{t, t+1\}$. Therefore,

$$\begin{aligned}
\hat{L}'_T - \hat{L}_T &= \sum_{s=1}^{T} \hat{\ell}'_s - \sum_{s=1}^{T} \hat{\ell}_s \\
&= (\hat{\ell}'_t + \hat{\ell}'_{t+1}) - (\hat{\ell}_t + \hat{\ell}_{t+1}) \\
&= p'_{t+1} - p_t \\
&= f(\eta_{t+1} \Delta'_t) - f(\eta_t \Delta_{t-1}) \\
&= f\big(\eta_{t+1}(\Delta'_{t-1} + 0)\big) - f(\eta_t \Delta_{t-1}) \\
&= f(\eta_{t+1} \Delta'_{t-1}) - f(\eta_t \Delta_{t-1}) \\
&= f(\eta_{t+1} \Delta_{t-1}) - f(\eta_t \Delta_{t-1}) \\
&\leq 0,
\end{aligned}$$

where the last inequality comes from the fact in Decreasing Hedge, $\eta_{t+1} < \eta_t$ and $f$ is an increasing function.

(b) In this case, the loss incurred by Decreasing Hedge for loss sequence $\ell_1, \ldots, \ell_T$ at rounds $t$ and $t+1$ can be expressed as

$$\hat{\ell}_t + \hat{\ell}_{t+1} = \underbrace{\begin{pmatrix} p_t \\ 1 - p_t \end{pmatrix} \cdot \begin{pmatrix} 0 \\ 0 \end{pmatrix}}_{\text{round } t} + \underbrace{\begin{pmatrix} p_{t+1} \\ 1 - p_{t+1} \end{pmatrix} \cdot \begin{pmatrix} 0 \\ 1 \end{pmatrix}}_{\text{round } t+1} = 1 - p_{t+1}.$$

Similarly the loss incurred by Decreasing Hedge for loss sequence $\ell'_1, \ldots, \ell'_T$ at rounds $t$ and $t+1$ can be written as

$$\hat{\ell}'_t + \hat{\ell}'_{t+1} = \underbrace{\begin{pmatrix} p'_t \\ 1 - p'_t \end{pmatrix} \cdot \begin{pmatrix} 0 \\ 1 \end{pmatrix}}_{\text{round } t} + \underbrace{\begin{pmatrix} p'_{t+1} \\ 1 - p'_{t+1} \end{pmatrix} \cdot \begin{pmatrix} 0 \\ 0 \end{pmatrix}}_{\text{round } t+1} = 1 - p'_t.$$

We know that Decreasing Hedge incurs the same loss at any round $s \in [T] \setminus \{t, t+1\}$; therefore,

$$\begin{aligned}
\hat{L}'_T - \hat{L}_T &= \sum_{s=1}^{T} \hat{\ell}'_s - \sum_{s=1}^{T} \hat{\ell}_s \\
&= (\hat{\ell}'_t + \hat{\ell}'_{t+1}) - (\hat{\ell}_t + \hat{\ell}_{t+1}) \\
&= (1 - p'_t) - (1 - p_{t+1}) \\
&= p_{t+1} - p'_t \\
&= f(\eta_{t+1} \Delta_t) - f(\eta_t \Delta'_{t-1}) \\
&= f\big(\eta_{t+1}(\Delta_{t-1} + 0)\big) - f(\eta_t \Delta'_{t-1}) \\
&= f(\eta_{t+1} \Delta_{t-1}) - f(\eta_t \Delta_{t-1}) \\
&\leq 0,
\end{aligned}$$

where the last inequality comes from the fact in Decreasing Hedge, $\eta_{t+1} < \eta_t$ and $f$ is an increasing function.

(c) In this case, as we assume that expert 1 is always the leader, $\Delta_s \geq 0$ for all $s$. Therefore, $\Delta_t = \Delta_{t-1} - 1 \geq 0$. The loss incurred by Decreasing Hedge for loss sequence $\ell_1, \ldots, \ell_T$ at rounds $t$ and $t+1$ can be expressed as

$$\hat{\ell}_t + \hat{\ell}_{t+1} = \underbrace{\begin{pmatrix} p_t \\ 1-p_t \end{pmatrix} \cdot \begin{pmatrix} 1 \\ 0 \end{pmatrix}}_{\text{round } t} + \underbrace{\begin{pmatrix} p_{t+1} \\ 1-p_{t+1} \end{pmatrix} \cdot \begin{pmatrix} 0 \\ 1 \end{pmatrix}}_{\text{round } t+1} = (p_t) + (1 - p_{t+1}).$$

Similarly, the loss incured by Decreasing Hedge for loss sequence $\ell'_1, \ldots, \ell'_T$ at rounds $t$ and $t+1$ can be written as

$$\hat{\ell}'_t + \hat{\ell}'_{t+1} = \underbrace{\begin{pmatrix} p'_t \\ 1-p'_t \end{pmatrix} \cdot \begin{pmatrix} 0 \\ 1 \end{pmatrix}}_{\text{round } t} + \underbrace{\begin{pmatrix} p'_{t+1} \\ 1-p'_{t+1} \end{pmatrix} \cdot \begin{pmatrix} 1 \\ 0 \end{pmatrix}}_{\text{round } t+1} = (1 - p'_t) + (p'_{t+1}).$$

We know that Decreasing Hedge incurs the same loss at any round $s \in [T] \setminus \{t, t+1\}$; therefore,

$$
\begin{aligned}
\hat{L}'_T - \hat{L}_T &= \sum_{s=1}^{T} \hat{\ell}'_s - \sum_{s=1}^{T} \hat{\ell}_s \\
&= \left( \hat{\ell}'_t + \hat{\ell}'_{t+1} \right) - \left( \hat{\ell}_t + \hat{\ell}_{t+1} \right) \\
&= (p'_{t+1} + p_{t+1}) - (p'_t + p_t) \\
&= \left( f(\eta_{t+1}\Delta'_t) + f(\eta_{t+1}\Delta_t) \right) - \left( f(\eta_t \Delta'_{t-1}) + f(\eta_t \Delta_{t-1}) \right) \\
&= \left( f(\eta_{t+1}\Delta'_t) + f(\eta_{t+1}\Delta_t) \right) - \left( 2 \cdot f(\eta_t \Delta_{t-1}) \right) \\
&= \left( f\left(\eta_{t+1}(\Delta_{t-1} + 1)\right) + f\left(\eta_{t+1}(\Delta_{t-1} - 1)\right) \right) - 2 \cdot f(\eta_t \Delta_{t-1}) \\
&= 2 \cdot \left( \frac{1}{2} f\left(\eta_{t+1}(\Delta_{t-1} + 1)\right) + \frac{1}{2} f\left(\eta_{t+1}(\Delta_{t-1} - 1)\right) \right) - 2 \cdot f(\eta_t \Delta_{t-1}) \\
&\leq 2 \cdot f\left( \frac{1}{2}\eta_{t+1}(\Delta_{t-1} + 1) + \frac{1}{2}\eta_{t+1}(\Delta_{t-1} - 1) \right) - 2 \cdot f(\eta_t \Delta_{t-1}) \\
&= 2 \cdot f(\eta_{t+1}\Delta_{t-1}) - 2 \cdot f(\eta_t \Delta_{t-1}) \\
&= 2 \cdot \left[ f(\eta_{t+1}\Delta_{t-1}) - f(\eta_t \Delta_{t-1}) \right] \\
&\leq 0,
\end{aligned}
$$

where the first inequality is Jensen's inequality applied for function $f$ which is concave for nonnegative domain and $\Delta_{t-1} - 1$ and $\Delta_{t-1} + 1$ are nonnegative. The second inequality comes from the fact that in Decreasing Hedge, $\eta_{t+1} < \eta_t$ and $f$ is an increasing function.

$\square$

Observe that by Lemma 16 we can always move $\binom{0}{1}$ to the earlier rounds and $\binom{1}{0}$ to the later rounds. If we apply this swapping rule until for no $(t, t+1)$ the conditions in the lemma hold, the resulting loss sequence will be of the form $\binom{0}{1}^a \binom{0}{0}^c \binom{1}{0}^b$. Note that $a \geq b$ as there is no leader change in the sequence.

## F.4 Regret is minimized when both experts have the same cumulative loss

So far we have shown that any loss sequence can be converted to a loss sequence with form $\left( \binom{0}{1}^a \binom{0}{0}^c \binom{1}{0}^b \right)$ where $a \geq b$ while having the same or even less regret for Decreasing Hedge. First we show that when $T$ is even, the number of zeros in the middle should be even.

**Lemma 17.** *If a sequence has the form* $\ell_1, \ldots, \ell_T = \left( \binom{0}{1}^{a-1} \binom{0}{1} \binom{0}{0}^{2k-1} \binom{1}{0}^b \right)$ *where $a \geq b$ and $T$ is even, then converting it to a loss sequence* $\ell'_1, \ldots, \ell'_T = \left( \binom{0}{1}^{a-1} \binom{0}{0} \binom{0}{0}^{2k-1} \binom{1}{0}^b \right)$ *decreases the regret.*

*Proof.* As $T = a + b + 2k - 1$, and $T$ is even, $a + b$ should be odd. Therefore, $a \neq b$. Which implies that $a > b$. Thus $a - 1 \geq b$. Therefore, the best expert in both loss sequences has the same loss:

$$\min_i L'_{i,T} = \min_i L_{i,T} = b.$$

Also, $\hat{\ell}'_t = \hat{\ell}_t$ for all $t < a$ as the loss sequences are the same for all rounds $t < a$.

In round $t = a$, Decreasing Hedge incurs some positive loss in $\ell_1, \ldots, \ell_T$ while it incurs zero loss in sequence $\ell'_1, \ldots, \ell'_T$. In other words, $\hat{\ell}'_a = 0 < \hat{\ell}_a$.

For rounds $t \in \{a + 1, \ldots, a + 2k - 1\}$, Decreasing Hedge incurs zero loss for both loss sequences $\ell_1, \ldots, \ell_T$ and $\ell'_1, \ldots, \ell'_T$. In other words, $\hat{\ell}'_t = \hat{\ell}_t = 0$.

Since $\Delta'_{t-1} = L'_{2,t-1} - L'_{1,t-1} = (L_{2,t-1} - 1) - L_{1,t-1} = \Delta_{t-1} - 1$ for any round $t \in \{a + (2k - 1) + 1, \ldots, a + (2k - 1) + b\}$,

$$\hat{\ell}'_t = \frac{1}{1 + \exp(-\eta_t \Delta'_{t-1})} = \frac{1}{1 + \exp\left(-\eta_t(\Delta_{t-1} - 1)\right)} \leq \frac{1}{1 + \exp(-\eta_t \Delta_{t-1})} = \hat{\ell}_t.$$

As a result, $\hat{\ell}'_t \leq \hat{\ell}_t$ for all $1 \leq t \leq T$. Therefore,

$$\mathcal{R}'(T) = \sum_{t=1}^{T} \hat{\ell}'_t - \min_i L'_{i,T} = \sum_{t=1}^{T} \hat{\ell}'_t - \min_i L_{i,T}$$

$$\leq \sum_{t=1}^{T} \hat{\ell}_t - \min_i L_{i,T} = \mathcal{R}(T).$$

$\square$

Now, we only consider loss sequences of form $\left(\binom{0}{1}^a \binom{0}{0}^c \binom{1}{0}^b\right)$ where $c$ is even. As $a + b + c = T$ and $T$ is even, $a + b = 2K$ is even.

Next, we will show in the following lemma that any loss sequence $\left(\binom{0}{1}^a \binom{0}{0}^c \binom{1}{0}^b\right)$ where $a + b = 2K$ and $a \geq b$ can be converted to a loss sequence $\left(\binom{0}{1}^K \binom{0}{0}^c \binom{1}{0}^K\right)$ while having the same or even less regret.

**Lemma 18.** *Among all loss sequences of form $\left(\binom{0}{1}^a \binom{0}{0}^c \binom{1}{0}^b\right)$ where $a + b = 2K$ and $a \geq b$, loss sequence $\left(\binom{0}{1}^K \binom{0}{0}^c \binom{1}{0}^K\right)$ has the least regret for Decreasing Hedge.*

*Proof.* Consider a sequence $\ell_1, \ldots, \ell_T = \left(\binom{0}{1}^a \binom{0}{0}^c \binom{1}{0}^b\right)$ where $a \neq b$. Therefore, $a > b$ which means that $a \geq b + 1$. If $a = b + 1$, then $a + b = 2b + 1 \neq 2K$. This means that $a \neq b + 1$. Therefore, $a \geq b + 2$.

Now, consider $\ell'_1, \ldots, \ell'_T = \left(\binom{0}{1}^{a-1} \binom{1}{0} \binom{0}{0}^c \binom{1}{0}^b\right)$. We know that the best expert in $\ell_1, \ldots, \ell_T$ incurs $L_{1,T} = b < a = L_{2,T}$ whereas the best expert in $\ell'_1, \ldots, \ell'_T$ incurs additional unit loss, since $L'_{1,T} = b + 1 \leq a - 1 = L'_{2,T}$. Therefore, $\min_i L'_{i,T} = b + 1 = \min_i L_{i,T} + 1$.

We now show that the cumulative loss of Decreasing Hedge for loss sequence $\ell'_1, \ldots, \ell'_T$ is at most 1 unit greater than cumulative loss for loss sequence $\ell_1, \ldots, \ell_T$. Therefore, $\mathcal{R}'(T) \leq \mathcal{R}(T)$.

For rounds $t < a$, Decreasing Hedge incurs the same loss for both loss sequences, i.e., $\hat{\ell}'_t = \hat{\ell}_t$ for all $t < a$. In round $t = a$, we have $\hat{\ell}'_a = \frac{1}{1+\exp(-\eta_a(a-1))} \geq 0$ and $\hat{\ell}_a = \frac{1}{1+\exp(\eta_a(a-1))} \geq 0$. Observe that $\hat{\ell}'_a + \hat{\ell}_a = \frac{1}{1+\exp(-\eta_a(a-1))} + \frac{1}{1+\exp(\eta_a(a-1))} = 1$. Therefore,

$$\hat{\ell}'_a - \hat{\ell}_a = (1 - \hat{\ell}_a) - \hat{\ell}_a = 1 - 2\hat{\ell}_a \leq 1.$$

As a result, in round $a$ Decreasing Hedge incurs at most 1 unit of additional loss in loss sequence $\ell'_1, \ldots, \ell'_T$ as compared to loss sequence $\ell_1, \ldots, \ell_T$.

For rounds $t \in \{a+1, \ldots, a+c\}$, Decreasing Hedge incurs zero loss in both loss sequences.

Note that since there is a change from $\binom{0}{1}$ to $\binom{1}{0}$ in round $a$, for any round $t \in \{a+1, \ldots, a+c+b\}$, we have $\Delta'_{t-1} = L'_{2,t-1} - L'_{1,t-1} = \Delta_{t-1} - 2$. Therefore, for any round $t$ in this period, Decreasing Hedge incurs less loss in $\ell'_1, \ldots, \ell'_T$ than $\ell_1, \ldots, \ell_T$:

$$\hat{\ell}'_t = \frac{1}{1 + \exp(-\eta_t \Delta'_{t-1})} = \frac{1}{1 + \exp\left(-\eta_t(\Delta_{t-1} - 2)\right)} \leq \frac{1}{1 + \exp(-\eta_t \Delta_{t-1})} = \hat{\ell}_t.$$

As a result, the cumulative loss of Decreasing Hedge in loss sequence $\ell'_1, \ldots, \ell'_T$ is at most 1 unit greater that Decreasing Hedge's cumulative loss in loss sequence $\ell_1, \ldots, \ell_T$, i.e.,

$$\sum_{t=1}^{T} \hat{\ell}'_t \leq 1 + \sum_{t=1}^{T} \hat{\ell}_t.$$

Therefore,

$$
\begin{aligned}
\mathcal{R}'(T) &= \sum_{t=1}^{T} \hat{\ell}'_t - \min_i L'_{i,T} \\
&= \sum_{t=1}^{T} \hat{\ell}'_t - (\min_i L_{i,T} + 1) \\
&\leq (1 + \sum_{t=1}^{T} \hat{\ell}_t) - (\min_i L_{i,T} + 1) \\
&= \sum_{t=1}^{T} \hat{\ell}_t - \min_i L_{i,T} = \mathcal{R}(T).
\end{aligned}
$$

Now by Lemma 16, we can move the loss vector $(1,0)$ in $\ell'_1, \ldots, \ell'_t$ in round $a$ to the right without increasing the regret. Therefore, the regret for $\ell''_1, \ldots, \ell''_t = \left(\binom{0}{1}^{a-1} \binom{0}{0}^c \binom{1}{0}^{b+1}\right)$ is no greater than the regret for $\ell_1, \ldots, \ell_T$. This means that when $a \geq b+1$, converting loss sequence $\left(\binom{0}{1}^a \binom{0}{0}^c \binom{1}{0}^b\right)$ to $\left(\binom{0}{1}^{a-1} \binom{0}{0}^c \binom{1}{0}^{b+1}\right)$ does not increase the regret. We can apply this rule many times to equalize the numbers $a$ and $b$. As $a + b = 2K$, this implies $a = b = K$ and the lemma follows. $\qquad\square$

### F.5 The optimum number of $(0,0)$ loss vectors in the middle

So far, we know that the best-case sequence has the form $\left(\binom{0}{1}^K \binom{0}{0}^c \binom{1}{0}^K\right)$ where $c$ is even. Is the existence of $\binom{0}{0}^c$ in the middle necessary to have the loss sequence with the least possible regret?

In order to examine this, we consider a simple modification to the loss sequence. Consider a loss sequence in which one $(0,0)$ is replaced with $(0,1)$ and another $(0,0)$ is replaced with $(1,0)$. Then by the swapping lemma (Lemma 16), we can move the $(1,0)$ to the later rounds and the $(0,1)$ to the earlier rounds without increasing the regret. If we denote the original sequence as $\ell_1, \ldots, \ell_T = \binom{0}{1}^a \binom{0}{0}^c \binom{1}{0}^b$, the modified loss sequence becomes $\ell'_1, \ldots, \ell'_T = \binom{0}{1}^{a+1} \binom{0}{0}^{c-2} \binom{1}{0}^{b+1}$.

**Lemma 19.** *Consider a loss sequence $\ell_1, \ldots, \ell_T = \binom{0}{1}^a \binom{0}{0}^c \binom{1}{0}^b$, where $c$ is even and $a \geq 4$. Then modifying loss sequence to $\ell'_1, \ldots, \ell'_T = \binom{0}{1}^{a+1} \binom{0}{0}^{c-2} \binom{1}{0}^{b+1}$ does not increase the regret if $c \geq 4$ and does increase the regret if $c \leq 3$.*

*Proof.* First of all, observe that both experts incur one additional unit loss in loss sequence $\ell'_1, \ldots, \ell'_T$ in comparison to $\ell_1, \ldots, \ell_T$. Therefore,

$$\max_i L'_{i,T} = \max_i L_{i,T} + 1.$$

Note that in any round $s$ except rounds $t = a+1$ and $t' = a+c$, we have $\hat{\ell}_s = \hat{\ell}'_s$.

On the other hand, for loss $\boldsymbol{\ell}'_1, \ldots, \boldsymbol{\ell}'_T$, in round $t = a+1$ and $t' = a+c$, Decreasing Hedge incurs loss

$$\hat{\ell}'_{a+1} = \frac{1}{1 + \exp(-\eta_t(a))},$$

$$\hat{\ell}'_{a+c} = \frac{1}{1 + \exp(\eta_{t'}(a+1))}.$$

while for $\boldsymbol{\ell}_1, \ldots, \boldsymbol{\ell}_T$, the algorithm does not incur any loss in those rounds (i.e. $\hat{\ell}_{a+1} + \hat{\ell}_{a+c} = 0$). Therefore, the difference in regret would be

$$\mathcal{R}'(T) - \mathcal{R}(T) = \frac{1}{1 + \exp(\eta_t(a))} + \frac{1}{1 + \exp(-\eta_{t'}(a+1))} - 1.$$

Let $f(x) := \frac{1}{1+\exp(-x)}$. Observe that $f(x) + f(-x) = 1$. Therefore,

$$\begin{aligned}
\mathcal{R}'(T) - \mathcal{R}(T) &= \frac{1}{1 + \exp(\eta_t(a))} + \frac{1}{1 + \exp(-\eta_{t'}(a+1))} - 1 \\
&= f(-\eta_t(a)) + f(\eta_{t'}(a+1)) - 1 \\
&= f\big(-\eta_t(a)\big) + f\big(\eta_{t'}(a+1)\big) - \Big(f\big(\eta_t(a)\big) + f\big(-\eta_t(a)\big)\Big) \\
&= f(\eta_{t'}(a+1)) - f(\eta_t(a)).
\end{aligned}$$

As $f$ is an increasing function, $f(\eta_{t'}(a+1)) - f(\eta_t(a)) \leq 0$ if and only if

$$\eta_{t'}(a+1) \leq \eta_t(a). \tag{16}$$

Therefore, if inequality (16) holds, then the replacement does not increase the regret. Substituting the learning rate, inequality (16) becomes

$$\frac{a+1}{\sqrt{a+c}} \leq \frac{a}{\sqrt{a+1}},$$

which is equivalent to

$$c \geq \frac{3a^2 + 3a + 1}{a^2} = 3 + \frac{3}{a} + \frac{1}{a^2}. \tag{17}$$

Note that in case $a \geq 4$ and $c \geq 4$, the inequlity $\frac{3}{a} + \frac{1}{a^2} < 1$ holds; therefore, (17) holds which means that the regret does not increase.

In the case $c \leq 3$, then (17) never holds which means that this replacement increases the regret. $\qquad \square$

Now, let us consider a loss sequence $\boldsymbol{\ell}_1, \ldots, \boldsymbol{\ell}_T$ of length $T = 2K \geq 16$ of the form $\binom{0}{1}^a \binom{0}{0}^c \binom{1}{0}^b$. If $a > b$, then we can and will use Lemma 18 to convert this sequence into a sequence for which $a = b$. The resulting sequence has the form $\binom{0}{1}^a \binom{0}{0}^c \binom{1}{0}^a$. We will now show how this sequence can be modified, without increasing the regret, to a sequence for which $c = 2$ (while maintaining the invariant $a = b$). There are two cases.

In the first case, the sequence is of the form $\binom{0}{1}^a \binom{0}{0}^c \binom{1}{0}^a$ where $a \geq 4$ and $c \geq 4$ (recall that $c$ must be even, and note that if $c = 2$, we are done). We may then apply Lemma 19 to decrease the number of $(0,0)$ vectors from the middle by two while maintaining the invariant $a = b$. As we have shown in Lemma 17 that $c$ is even, the last time where we can apply this lemma is when $c = 4$ and the resulting sequence is of the form $\binom{0}{1}^{a + \frac{c-2}{2}} \binom{0}{0}^2 \binom{1}{0}^{\frac{c-2}{2} + a}$

In the second case, we have $a \in \{1, 2, 3\}$. Since $a + b + c = T \geq 16$ and as we may assume that $a = b$, it follows that $c \geq 10$. In this case, inequality (17) holds, which implies that modifying the loss sequence to $\boldsymbol{\ell}'_1, \ldots, \boldsymbol{\ell}'_T = \binom{0}{1}^{a+1} \binom{0}{0}^{c-2} \binom{1}{0}^{a+1}$ does not increase the regret. After applying this modification at most three times, the resulting loss sequence will be of the form $\boldsymbol{\ell}''_1, \ldots, \boldsymbol{\ell}''_T =$

$\binom{0}{1}^a \binom{0}{0}^c \binom{1}{0}^a$ where $a \geq 4$. We are now in the first case and so again can arrive at a sequence with two $(0,0)$ vectors in the middle.

Observe that $\binom{0}{1}^{a+\frac{c-2}{2}} \binom{0}{0}^2 \binom{1}{0}^{\frac{c-2}{2}+a}$ is of the form $\binom{0}{1}^K \binom{0}{0}^2 \binom{1}{0}^K$. This form coincides with the canonical best-case sequence we mentioned earlier.

### F.6 Bounding the regret

So far, we have shown that among all loss sequence of size $T = 2K + 2 \geq 16$, the loss sequence $\left( \binom{0}{1}^K \binom{0}{0}^2 \binom{1}{0}^K \right)$ yields the minimum regret for Decreasing Hedge. We will now give upper and lower bounds on the regret for this sequence. The algorithm's loss for this sequence is

$$\hat{L}_T = \sum_{t=1}^T \hat{\ell}_t = \hat{\ell}_1 + \underbrace{\sum_{t=2}^K (\hat{\ell}_t)}_{\binom{0}{1}^K} + \underbrace{\hat{\ell}_{K+1} + \hat{\ell}_{K+2}}_{\binom{0}{0}^2} + \underbrace{\hat{\ell}_{K+3} + \sum_{t=2}^K \hat{\ell}_{2K+4-t}}_{\binom{1}{0}^K}$$

$$= \hat{\ell}_1 + \hat{\ell}_{K+3} + \sum_{t=2}^K \hat{\ell}_t + \hat{\ell}_{2K+4-t}$$

$$= \frac{1}{2} + \frac{1}{1 + \exp(-\eta_{K+3}(K))} + \sum_{t=2}^K \left( \frac{1}{1 + \exp\left(\eta_t(t-1)\right)} + \frac{1}{1 + \exp\left(-\eta_{2K+4-t}(t-1)\right)} \right).$$

Therefore,

$$\mathcal{R}(T) = \hat{L}_T - \min_i L_{i,T}$$

$$= \hat{L}_T - K$$

$$= \sum_{t=2}^K \left( \frac{1}{1 + \exp\left(\eta_t(t-1)\right)} + \frac{1}{1 + \exp\left(-\eta_{2K+4-t}(t-1)\right)} \right)$$
$$+ \frac{1}{2} + \frac{1}{1 + \exp(-\eta_{K+3}(K))} - K$$

$$= \sum_{t=2}^K \left( \frac{1}{1 + \exp\left(\eta_t(t-1)\right)} + \frac{1}{1 + \exp\left(-\eta_{2K+4-t}(t-1)\right)} - 1 \right) + \sum_{t=2}^K 1$$
$$+ \frac{1}{2} + \frac{1}{1 + \exp(-\eta_{K+3}(K))} - K$$

$$= \sum_{t=2}^K \left( \frac{1}{1 + \exp\left(\eta_t(t-1)\right)} + \frac{1}{1 + \exp\left(-\eta_{2K+4-t}(t-1)\right)} - 1 \right)$$
$$+ \frac{1}{2} + \frac{1}{1 + \exp(-\eta_{K+3}(K))} - 1.$$

As we know that $\frac{1}{1+\exp\left(-\eta_{2K+4-t}(t-1)\right)} + \frac{1}{1+\exp\left(+\eta_{2K+4-t}(t-1)\right)} = 1$, it follows that

$$\frac{1}{1 + \exp\left(\eta_t(t-1)\right)} + \frac{1}{1 + \exp\left(-\eta_{2K+4-t}(t-1)\right)} - 1 = \frac{1}{1 + \exp\left(\eta_t(t-1)\right)} - \frac{1}{1 + \exp\left(\eta_{2K+4-t}(t-1)\right)}.$$

Hence, we can rewrite the regret as

$$\mathcal{R}(T) = \underbrace{\sum_{t=2}^K \frac{1}{1 + \exp\left(\eta_t(t-1)\right)}}_{A = \text{First term}} - \underbrace{\sum_{t=2}^K \frac{1}{1 + \exp\left(\eta_{2K+4-t}(t-1)\right)}}_{B = \text{Second term}} + \underbrace{\frac{1}{2} + \frac{1}{1 + \exp(-\eta_{K+3}(K))} - 1}_{C = \text{Third term}}.$$

We will give upper and lower bounds for each term separately. Note that $K = \frac{T-2}{2}$. In the development of the bounds below, we assume that $T$ is sufficiently large. It suffices to have $T \geq 16$.

Now, for the A (First term), we have

$$
\begin{aligned}
0 \le \mathrm{A} = \sum_{t=2}^{K} \frac{1}{1 + \exp(\eta_t(t-1))} &= \sum_{t=2}^{K} \frac{1}{1 + \exp(\sqrt{t} - \frac{1}{\sqrt{t}})} \\
&\le \sum_{t=2}^{K} \frac{1}{1 + \exp(\frac{1}{2}\sqrt{t})} \le \sum_{t=2}^{K} \frac{1}{\exp(\frac{1}{2}\sqrt{t})} \\
&\le \int_{1}^{K} \frac{1}{\exp(\frac{1}{2}\sqrt{x})} \, dx \\
&= (-4\sqrt{K} - 8)\exp(-0.5\sqrt{K}) - (-4\sqrt{1} - 8)\exp(-0.5\sqrt{1}) \\
&\le (4\sqrt{1} + 8)\exp(-0.5\sqrt{1}) \\
&= \frac{12}{\sqrt{e}}.
\end{aligned}
$$

To upper bound B (Second term), we divide the $T$ rounds into $\lceil \sqrt{T} \rceil$ intervals. We then show that sum of the contributions of the intervals is upper bounded by sum of a geometric series that converge to a constant value.

The second term can be written as the sum of $\lceil \sqrt{T} \rceil$ intervals as

$$
\begin{aligned}
\sum_{t=2}^{K} \frac{1}{1 + \exp\left(\eta_{2K+4-t}(t-1)\right)} &= \sum_{t=2}^{\frac{T}{2}-1} \frac{1}{1 + \exp\left(\eta_{T+2-t}(t-1)\right)} \\
&\le \sum_{t=1}^{T} \frac{1}{1 + \exp\left(\frac{t-1}{\sqrt{T+2-t}}\right)} \\
&= \sum_{k=0}^{\lceil\sqrt{T}\rceil-1} \sum_{t=k\lceil\sqrt{T}\rceil+1}^{k\lceil\sqrt{T}\rceil+\lceil\sqrt{T}\rceil} \frac{1}{1 + \exp\left(\frac{t-1}{\sqrt{T+2-t}}\right)} \\
&\le \sum_{k=0}^{\lceil\sqrt{T}\rceil-1} \sum_{t=k\lceil\sqrt{T}\rceil+1}^{k\lceil\sqrt{T}\rceil+\lceil\sqrt{T}\rceil} \frac{1}{1 + \exp(\frac{k\sqrt{T}}{\sqrt{T+2}})} \\
&\le \lceil\sqrt{T}\rceil \sum_{k=0}^{\lceil\sqrt{T}\rceil-1} \frac{1}{1 + \exp(\frac{k\sqrt{T}}{2+\sqrt{T}})}.
\end{aligned}
$$

The term $\sum_{k=0}^{\lceil\sqrt{T}\rceil-1} \dfrac{1}{1+\exp(\frac{k\sqrt{T}}{2+\sqrt{T}})}$ can be upper bounded as

$$\sum_{k=0}^{\lceil\sqrt{T}\rceil-1} \frac{1}{1+\exp(\frac{k\sqrt{T}}{2+\sqrt{T}})} = \sum_{k=0}^{\lceil\sqrt{T}\rceil-1} \frac{1}{1+\exp\left(k-\frac{2k}{2+\sqrt{T}}\right)} \le \sum_{k=0}^{\lceil\sqrt{T}\rceil-1} \frac{1}{1+\exp(k-2)}$$

$$\le \sum_{k=0}^{\lceil\sqrt{T}\rceil-1} \frac{1}{\exp(k-2)}$$

$$= e^2 \sum_{k=0}^{\lceil\sqrt{T}\rceil-1} \left(\frac{1}{e}\right)^k$$

$$\le e^2 \left(\frac{1-\frac{1}{e}^{\lceil\sqrt{T}\rceil}}{1-\frac{1}{e}}\right)$$

$$\le \frac{e^2}{1-\frac{1}{e}}.$$

As a result,

$$\sum_{t=2}^{K} \frac{1}{1+\exp\left(\eta_{2K+4-t}(t-1)\right)} \le \left\lceil\sqrt{T}\right\rceil \cdot \sum_{k=0}^{\lceil\sqrt{T}\rceil-1} \frac{1}{1+\exp(\frac{k\sqrt{T}}{2+\sqrt{T}})} \le \frac{e^2}{1-\frac{1}{e}} \cdot \left\lceil\sqrt{T}\right\rceil.$$

Moreover, B (Second term) can be lower bounded as follows:

$$\sum_{t=2}^{K} \frac{1}{1+\exp\left(\eta_{2K+4-t}(t-1)\right)} = \sum_{t=2}^{K} \frac{1}{1+\exp\left(\eta_{T+2-t}(t-1)\right)}$$

$$= \sum_{t=2}^{\frac{T}{2}-1} \frac{1}{1+\exp(\frac{t-1}{\sqrt{T+2-t}})}$$

$$\ge \sum_{t=2}^{\lceil\sqrt{T}\rceil+1} \frac{1}{1+\exp(\frac{t-1}{\sqrt{T+2-t}})}$$

$$\ge \sum_{t=2}^{\lceil\sqrt{T}\rceil+1} \frac{1}{1+\exp\left(\frac{\lceil\sqrt{T}\rceil}{\sqrt{T+2-t}}\right)}$$

$$\ge \sum_{t=2}^{\lceil\sqrt{T}\rceil+1} \frac{1}{1+\exp\left(\frac{\sqrt{T}}{\sqrt{T+1-\lceil\sqrt{T}\rceil}}\right)}$$

$$\ge \sum_{t=2}^{\lceil\sqrt{T}\rceil+1} \frac{1}{1+\exp\left(\frac{\lceil\sqrt{T}\rceil}{\sqrt{\frac{1}{2}\lceil\sqrt{T}\rceil}}\right)}$$

$$= \sum_{t=2}^{\lceil\sqrt{T}\rceil+1} \frac{1}{1+\exp(\sqrt{2})}$$

$$\ge \frac{\sqrt{T}}{1+\exp(\sqrt{2})}.$$

Therefore,

$$\frac{1}{1+e^{\sqrt{2}}}\sqrt{T} \le B = \sum_{t=2}^{K} \frac{1}{1+\exp\left(\eta_{2K+4-t}(t-1)\right)} \le \frac{e^2}{(1-\frac{1}{e})}\left\lceil\sqrt{T}\right\rceil.$$

For the C (Third term), as we know $0 \le \hat{\ell}_{K+3} = \frac{1}{1+\exp(-\eta_{K+3}(K))} \le 1$, it follows that

$$-\frac{1}{2} \le C \le \frac{1}{2}.$$

As a result, we have

$$-\frac{e^2}{(1-\frac{1}{e})}\cdot\left\lceil\sqrt{T}\right\rceil - \frac{1}{2} \le A - B + C \le \frac{12}{\sqrt{e}} + \frac{1}{2} - \frac{1}{1+e^{\sqrt{2}}}\sqrt{T},$$

which for suitably defined constants is

$$-c_1\sqrt{T} + c_2 \le \mathcal{R}(T) \le c_3 - c_4\sqrt{T}.$$

Therefore,

$$\mathcal{R}(T) = -\Theta(\sqrt{T}).$$