# OpenReview forum: "Best-case lower bounds in online learning"
_NeurIPS.cc/2021/Conference — NeurIPS 2021 Poster_

### Official Review · Reviewer_3Dws · 2021-07-13

**Rating:** 7
**Confidence:** 4

**Summary:**

Inspired by applications of fair online learning, the authors initiate the study of *lower bounds* on the regret of OCO algorithms. They provide a general lower bound for FTRL algorithms and show the implications for fixed regularizers and for some special kinds of time-varying regularizers (such as fixed regularizers with time-varying learning rates or regularizers dependent on the past losses). They also provide regret lower bounds on AdaGrad-like algorithms derived from the FTRL framework, and show how linearizing the losses can yield to algorithms with potentially $-\Omega(T)$ regret in $T$ rounds. They also explicitly build the best-case losses for the experts problem with 2 experts, giving sharp bounds for this special case;

Their main application is to fairness for decision theoretic online learning (experts' problem), strengthening previous fairness bounds. They show a way to get fairness guarantees without needing to know the size of the groups, and extending the fairness notion in online learning in a way that is independent of the length of the game.

**Limitations And Societal Impact:**

Yes.

**Main Review:**

# Strengths
- A solid theoretical paper, with a novel perspective on online optimization/learning and initiating a possibly fruitful direction of research;
- A solid contribution to fairness for DTOL, analyzing a natural fair algorithm that does not need to know the size of the groups ahead of time, and a good discussion on how the current notion of fairness from previous work is highly dependent on the time horizon;
- Very good characterization of the connection between fairness and lower-bounds on the regret;
- An example of an algorithm with a sublinear regret upper bound but with a linear regret lower bound, leaving open interesting questions about the characterization of algorithms that suffer from linear best-case lower-bounds.

# Weaknesses
- Implicit assumption on boundedness of the feasible set in the more general analysis of FTRL and some hidden assumptions on the regularizers that are not mentioned anywhere (and can even render the results false), details later;
- A lack of motivation (very interesting theoretically, but the broad NeurIPS audience might have a hard time understanding the motivation of the work);
- Regret bounds for the AdaGrad-like algorithms have an (apparently) extra dependence on a bound on the gradients that is not present in the original AdaGrad algorithm;
- The result for 2 experts, although exact, seems to lack a bit of discussion/motivation;

# Summary of the decision

A solid theoretical contribution with strong connections to fairness. So I consider it to be a good paper (7), although not higher since the motivation and applications, although interesting, are still narrow. I still expect comments of the authors on some of my points (mainly (3)) to maintain my score, but I believe none of these are serious problems.

# Detailed discussion

1) I really believe this paper is a solid theoretical contribution and initiates a direction of research that, beyond the clear applications to fairness already discussed, can have connections to stability of online learning algorithms and give some intuition on algorithms that can perform "extraordinarily" well in practice. Of course, it is not a completely new field, but it is an interesting perspective. However, I think the authors should make an extra-effort to sell this theoretical work to a broader community. NeurIPS has a vast audience, and having a better discussion of why studying these best-case lower-bounds is interesting. Even the gist of the discussion on fairness is deferred to the appendix. Bringing some of that discussion to the main body might already give a broader and strong motivation for the work. I really enjoyed the discussion from the appendix.

2) On Section 3 the authors state that $\mathcal{W}$ is a closed set, but do not mention necessarily that it is bounded. However, in the definition of regret itself the supremum may easily be infinite if $\mathcal{W}$ is the whole space. On Theorem 1 something similar happens, where the authors assume that there is a optimal action in hindsight, which might not be true (again, take the feasible set as being the whole space). A more fine-grained analysis shows that the regret against some comparison point is dependent on the regularizer at this comparison point, so boundedness is not necessary for (at least some) results to follow. Since all the application in the paper only look at cases where the the feasible region is bounded, this is not a serious problem, but clearly stating these assumptions would be good.

3) Another problem is that Theorem 1 requires the regularizers to be non-negative (not mentioned) and strongly convex (mentioned, but only inside the theorem, so it is not clear that the regularizers in the paper are assumed to be strongly convex throughout). These conditions **should** be explicitly stated and enforced, or at least the the statement of the FTRL regret bound should be adapted (subtracting the infimum of the last regularizer in the right hand side might be enough, but I haven't checked carefully). These assumptions on the regularizers and/or boundedness of $\mathcal{W}$ would help a lot in making the analysis in Section 2 less bumpy (without boundedness or strong-covexity, taking the infimum of the regularizer over the feasible set can lead to $-\infty$). In fact, if I'm not mistaken the authors themselves stumble onto these problems: on line 172 the authors *EXPLICITLY* use that the regularizer is non-positve on the entire feasible set. Applying this same regularizer to Theorem 1 we would have that the sup is also non-positive and we could get a logarithmic regret upper bound, and this is clearly false. Thus, this can make the readers very confused about which are the conditions required on the regularizers at each step for the results to hold. My suggestion is to adapt Theorem 1 for the case where the regularizers may be negative, but still strongly convex (and, thus, lower-bounded). But above all else, **make the assumptions on the regularizers clear**. (Another lesser point, the iterate from (1) might not be unique if the regularizers are not strictly convex).

4) On Sec 4.2 the authors should be a bit more explicit about the difference between AdaGrad and your algorithm (linear approx vs whole function). But something that is bugging me that I did not have the time to verify carefully is the dependence on bounds of the p-norms of the gradient in the regret bounds. I believe this FTRL-AdaGrad algorithm should get similar bounds to the original linearized version (in particular, if the functions are linear these algorithms match exactly). Although this is not the focus of the paper, this extra dependency is a bit unsettling, and I couldn't verify the proof since it is not provided in the appendix. So I suggest you to give the proof in the appendix, if time allows. But beyond that, I think a careful analysis (possibly using the strong FTRL Lemma from McMahan) should yield bounds without the extra dependency on the bounds of the gradients. I'd really love the thoughts of the authors on that.

5) The result on identifying exactly the best sequence for two experts is interesting, but I did not get much of the motivation for it. The discussion on why it shows that the previous result was tight is not clear (do you mean asymptotically only? Because the constants do not seem to match). But this part I might have simply misunderstood and is not a serious problem

**Time Spent Reviewing:**

6

---

> ### Author Response · Authors · 2021-08-09
> **Authors' Response**
>
> We would like to thank the reviewer for the thorough review and constructive feedback. Next we provide more detailed answers to the  comments raised.
>
> R: “[..] having a better discussion of why studying these best-case lower bounds is interesting[..]”
>
> A: Thank you for your comment on how to increase the reach of our work; this certainly is desirable. We will try to re-work the main text discussion of fairness by drawing from the detailed development that appears in the fairness appendix. If the reviewer noted particular things appearing in the fairness appendix which they would like to see in the main text, we would be very grateful if you could highlight a few of those items. If not, we of course are equipped to make a good attempt at this. Going beyond fairness, we will more heavily stress that the concept of best-case lower bounds may point out a new perspective on adaptivity of online learning algorithms. For example, our results in Section 5 point out that there exist instances with negative linear regret for linearized FTRL methods. We believe a more general characterization of such instances for algorithms observed to perform extremely well in practice, such as AdaGrad, may shed some light on this adaptivity. In particular, we know AdaGrad outperforms online gradient descent by its re-scaling properties, but there may be even additional features of the losses that permit outperforming the standard worst case regret analysis.
>
> R: “On Section 3, the authors state W is a closed set, but do not mention necessarily that it is bounded.[..]”
>
> A: This was an unintended omission, and we apologize for it. Clearly, most standard results in online learning and OCO require boundedness, and our results implicitly do so. We will make sure to correct this in the final version. Also, after explicitly assuming boundedness, the optimal action in hindsight will exist (and other infs and sups will be achieved), from compactness of the domain and continuity of the losses.
>
> R: “Another problem is that Theorem 1 requires the regularizers to be non-negative[..]”
>
> A: As far as we can see, the restrictions mentioned by the reviewer only apply to upper bounds on the regret. Notice that our best-case lower bounds do not need strong convexity assumptions. In fact, we prefer to keep the statements that are relevant for the paper contributions as general as possible: for instance, this allows us to argue about FTL. However, the reviewer is absolutely right that the current statement of Theorem 1 is missing a nonnegativity assumption on the regularizer. In the final version, we will change Theorem 1 to have two parts. The first part will be the currently stated version but with explicit mention of the nonnegativity assumption on the regularizers. The second part, which will not require nonnegativity of the regularizers, will follow the reviewer’s suggestion of modifying each regularizer by subtracting the infimum of the regularizer from the final iterate; this does yield a valid statement provided that we assume that the sequence of (unshifted) regularizers is pointwise non-increasing (i.e., for each $w \in \mathcal{W}$, the sequence $(\Phi_t(w))_t$ is non-increasing), and so for this part we will include this assumption. As Theorem 1 is meant to primarily be exemplary, this suffices to capture the key examples (e.g., Decreasing Hedge).
>
> R: “On Sec 4.2 the authors should be a bit more explicit about the difference between AdaGrad and your algorithm (linear approx vs whole function).[..]”
>
> A: We believe the reviewer may be mixing up two different aspects of adaptive gradient algorithms. We argue next that the offsets $D_2^2M_{\infty}/D_{\infty}$ and $D_2M_2/2$ (for diagonal and matrix scaling, respectively) that appear in our upper bounds are inherent to the algorithm we study. Our adaptive gradient FTRL algorithm, in its linear counterpart, corresponds to the so-called dual averaging method (or primal-dual, as called in update (3) of Duchi, Hazan, Singer). The standard AdaGrad algorithm, which does not include the offsets, corresponds to the so-called proximal variant (see update (4) in Duchi, Hazan, Singer). This difference is subtle, but crucial, regarding the offset terms: as discussed in Duchi, Hazan, Singer (right after Lemma 4 here https://www.jmlr.org/papers/volume12/duchi11a/duchi11a.pdf), the proximal variant can choose $\delta=0$ for the quadratic form, whereas the dual averaging counterpart requires $\delta \geq \max_t \Vert g_t \Vert_{\infty}$ (these discussions also appear in McMahan 2017: see Sections 3.5 and 3.7 of that paper). This is exactly the source of discrepancy, and why the offsets appear. Finally, these requirements on $\delta$ apply equally to either the linearized or non-linearized versions of the algorithms. We would like in future work to address the proximal variant of AdaGrad (which is closely related to online mirror-descent (OMD)), but we currently do not have an analog of our Theorem 2 for OMD, and in fact this variant may require completely different techniques. Proving a best-case lower bound for the linearized regret for OMD is actually quite straightforward, but keeping track of the (non-linearized) regret is currently out of reach, to the best of our efforts.
>
> To conclude, we would like to emphasize that the offsets arising in upper bounds do not appear in our worst-case lower bounds, which in our view only make our contributions stronger. Given that this is indeed a subtle matter, we will include a more detailed algorithm description and clarification on the nature of the offsets in the final version of the paper. We will also include in the supplementary material a more explicit derivation of Theorem 3 in our paper, based on direct application of McMahan’s and Duchi, Hazan, Singer’s papers.
>
> R:”The result on identifying the best case sequence for two experts is interesting, but I did not get much of the motivation for it.”
>
> A: The motivation for identifying the best-case sequence is three-fold. First, we wanted to demonstrate that our best-case lower bound for this setting (obtainable from our general result’s specialization in Section 4) is tight, i.e., has the right order (although we can’t yet attest to sharpness, i.e., having even the right constant factors). Second, pinpointing the best-case sequence sheds insight on the structure of loss sequences for which Decreasing Hedge obtains the lowest regret. For instance, consider a loss sequence $\ell_{1:T} = (0,1)^{T/3} (1,0)^{T/3} (0,1)^{T/3}$ and $\ell’_{1:T} = (0,1)^{2T/3} (0,1)^{T/3}$. It is not obvious for which loss sequence Decreasing Hedge performs better. However, as the cumulative loss of the best expert is the same in both sequences, we can use our lemmas to show that Decreasing Hedge performs better in $\ell’_{1:T}$ than in $\ell_{1:T}$. Notice that the best shifting sequence of experts for $\ell_{1:T}$ has two shifts while the best shifting sequence of experts for $\ell’_{1:T}$ has only one shift. More generally, our results from this section show that Decreasing Hedge has the lowest regret for loss sequences for which the best shifting sequence of experts has precisely one shift. One possible research direction would be to see if this pattern holds more generally, for other FTRL-type algorithms. Third, we have found few instances of this style of proof technique in the literature (outside of those papers we cited) and hoped to further the knowledge and application of this technique.

---

### Official Review · Reviewer_M8jx · 2021-07-15

**Rating:** 7
**Confidence:** 3

**Summary:**

This paper studies the best-case lower bound, which characterizes the minimum regret that a specific algorithm can achieve in the most benign environments. The authors provide the best-case lower bound for the general FTRL algorithm and show that the bound can be applied for many instances, including Hedge with non-increasing step size, adaptive gradient FTRL. The best-case lower bound for the Hedge algorithm can be used to prove the group fairness in the prediction with expert advice setting. Case studies on the linearized FTRL and Hedge with two experts are also provided.

**Ethical Concerns:**

No ethical concerns.

**Main Review:**

This paper provides a notion of best-case lower bound and analysis for the adaptive FTRL algorithm. The best-case bound is novel to me, and the analysis is simple yet general to be applied for many specific cases. In Section 3, the authors further show that the analysis can be applied to Hedge with time-varying step sizes, which can finally lead to group fairness in prediction with expert advice setting without knowing the time horizon in advance.

Although the analysis best-case lower bound can be used to ensure the group fairness for Interleaved Hedge algorithm, I still quite catch the significance of this study in a more general OCO setting. It seems that the best-case bound is neither a good measure for the performance of algorithms nor reflects the hardness of the problem. I think it would be nice for the author to illustrate more the significance of the study in a more general case.  For example, what would imply if we could obtain a best-case lower bound for adaptive gradient FTRL (Section 4.2)?

Besides, I am confused by the title of Section 5. Why the $-\\Omega(T)$ best-case lower bound of linear FTRL is called "negative result." Low regret seems good news for the OCO problem. Perhaps, the result is negative for achieving group fairness. But, the case study in Section 5 is presented in the OCO setting, and there is no concern about fairness. The confusion might be related to the main concern of the significance of the best-case lower bound in the OCO setting,

Overall, although the paper shows that the proposed best-case lower bound can be used to achieve group fairness of Hedge, its significance in the OCO setting is unclear. It would be better if the authors can illustrate more about some more applications of the best-case lower bound.

=====update=====

Thank you for the response. After reading other reviews and the author's feedback, my concerns are largely addressed. I would like to raise my score to a positive evaluation (7).

My original concern is that the motivation for studying the best-case lower bound in general OCO problem is unclear. By point (1),(2) of the first answer, the authors show that the best-case lower bound can be used to achieve group fairness beyond the expert advice setting. Although group fairness is an important notion, the extension could be limited to a narrow scope of audience. This concern is somewhat addressed by the third point, where authors have shown that the best-case lower bound can be used to show some negative side of the adaptive algorithms. For example, the adaptive FTRL does not perform that well in extraordinarily benign environments (it seems that the adaptive FTRL even is even inferior to the linearized FTRL in such a case). So, I am convinced that the best-lower bound can indeed provide a unique view to understand the adaptivity of the online learning algorithms.

Regarding the second question, I think the confusion might come from a lack of further discussion about the best-case lower bound. For the fairness problem, the less negative regret bound means better results. But, that is not the case for regret minimization. So, I think it would be better to provide a more clear discussion about the signification of the best-case lower bound for both scenarios (especially for the regret minimziation part).

**Time Spent Reviewing:**

5-6 hours

---

> ### Author Response · Authors · 2021-08-09
> **Authors' Response**
>
> We would like to thank the reviewer for the feedback. Next, we provide specific answers:
>
> R: “[..]  I still quite catch the significance of this study in a more general OCO setting.[..]”
>
>
> A: You are absolutely correct that a best-case lower bound is not, by itself, a good measure for the performance of a learning algorithm. Indeed, an algorithm could always (at least for a fixed time horizon) be rigged to have negative linear best-case regret while having linear worst-case regret, and many would consider such an algorithm to be a bad one. In addition, since best-case lower bounds are algorithm-specific, we agree that they do not reflect the hardness of a problem. We offer the following motivations for best-case lower bounds:
>
> (1) In the general OCO setting, an envy-related notion of group fairness might require that, when running a separate copy of an algorithm on each group, no copy of the algorithm has performance which significantly (say, up to $O(\sqrt{T})$) differs from the performance of the best expert for that group. Note that this is different from requiring similar cumulative loss for each group, as we are not assuming that each expert (or, in particular, the best expert for each group) has the same cumulative loss across groups. Instead, we ask for similar regret for each group (again, up to $\pm O(\sqrt{T})$).
>
> (2) Note that the best-case lower bound for adaptive gradient FTRL includes, as a special case, the use of this method with linear losses; in this case, OCO of course reduces to OLO. Now, supposing that the convex set has a finite number of vertices, it may be natural to assume that each vertex satisfies a notion of anytime group fairness that was developed in Section 4.1 of our paper, and from linearity, each point of the convex decision set also satisfies this notion of group fairness. Consequently, our fairness results automatically apply to adaptive FTRL, including the special case of adaptive gradient FTRL (which, since we are talking about OLO, corresponds to AdaGrad).
>
> (3) Also keeping in mind adaptive gradient FTRL (but more as a specific example of a highly adaptive algorithm, in a certain sense), the purpose of Section 4.2 is that even though this algorithm is adaptive in terms of gradients, it fails to be adaptive in the sense of being able to compete with a stronger comparator (like that used in a shifting regret). Hence, while it is adaptive, its adaptivity is limited. Depending on one’s perspective, this is a negative result.
>
> Also, although we do not yet have something rigorous to say about the following point yet, we do wish to mention it: in statistical decision theory, there is a notion of equalizer strategies which always obtain their worst-case regret, regardless of which probability distribution (from some fixed family) is chosen by Nature. See e.g. [(Grünwald and Dawid, AoS 2004)](https://projecteuclid.org/journals/annals-of-statistics/volume-32/issue-4/Game-theory-maximum-entropy-minimum-discrepancy-and-robust-Bayesian-decision/10.1214/009053604000000553.full) or [(Berger, 1985)](https://www.springer.com/gp/book/9780387960982). We wonder if something analogous can eventually be established in online learning, where algorithms that are minimax optimal never have regret whose order (ignoring the sign) is ever higher than the minimax rate. Note that our example of linearized FTRL is not a counterexample, as we employed the squared loss, in which case the minimax regret is of order $\log T$ rather than $\sqrt{T}$. Had we used non-linearized (i.e., standard) FTRL in Section 5, then the minimax optimal regret would be achieved and our best-case lower bounds show that the algorithm never obtains regret that is (ignoring the sign) higher than this, in order.
>
> R: “Besides, I am confused by the title in Section 5.”
>
>
> A: It is a matter of perspective for why we said “negative result” in the title of Section 5, and in hindsight, we feel similarly to the reviewer that the name is too tied to a particular frame. We will plan to change this title/framing. Indeed, when not considering fairness, it is quite a positive/good result to be able to sometimes achieve -$\Omega(T)$ regret while still enjoying $O(\sqrt{T})$ worst-case regret.

---

### Official Review · Reviewer_fZb1 · 2021-07-16

**Rating:** 7
**Confidence:** 2

**Summary:**

The paper establishes lower bounds on the best-case regret of the classical FTRL algorithm in online convex optimization. Specifically, the FTRL algorithm is considered in the general form where the regularizer can change over time (e.g., due to a time-varying learning rate). In the case of a fixed regularizer, it is known from earlier work that the regret is always non-negative, i.e., the algorithm never performs better than the best static expert. For time-varying regularizers, this is no longer the case, meaning that the best-case regret can be negative. The paper proves a simple generic lower bound theorem and applies it to several standard settings (Decreasing Hedge; a standard timeless method for decision-theoretic online learning (DTOL); an adaptive gradient version of FTRL), where it is found that the best-case lower bounds are roughly symmetrical to the known worst-case upper bounds on the regret (i.e., if A is a worst-case upper bound, then -O(A) is a best-case lower bound). In other words, these algorithms cannot outperform the best static expert by much, for any input. For DTOL with 2 experts, they show their lower bound to be tight.

For a different setting of the linearized FTRL algorithm, they show that an analogue result does not hold: Although the known regret upper bound is O(sqrt(T)), the best-case regret can be as low as -Omega(T).

An application and the original motivation for the work is in the setting of "group fairness" for DTOL: Here, time steps are partitioned into groups, and the algorithm knows which group the current time step belongs to. It is assumed that for each expert, the average loss per time step is balanced across the groups. Then recent work of Blum et al shows that the learning algorithm also has comparable loss in each group. This assumes that the time horizon T is known. This paper allows extending this result to an unknown time horizon, and shows an "anytime" extension of the result: If the assumption is strengthened so that for each expert, the losses are balanced across the groups not only after the last time step but also on every prefix of the input, then the learning algorithm also has balanced losses per group at all timesteps.

**Ethical Concerns:**

No concerns.

**Limitations And Societal Impact:**

Yes.

**Main Review:**

The results seem to be an interesting generalization of previous work by allowing time-varying regularizers. In particular, they show limitations on outperforming the best static expert in several standard settings (as would be necessary in a "shifting regret" settings) and the extension of previous group fairness results seem relevant. However, someone more familiar with the related work might be a better judge of the overall significance then me.

The paper is well-written and ideas easy to understand.

Minor comments:
- On pages 3-4, there are at least superscripts of w that should be subscripts.
- line 175: factor 2 is missing eta_t
- In the math after line 206 and after 209, it seems the terms on the LHS should be divided by the number of time steps in the group.
- line 365: "me" -> "be"

**Time Spent Reviewing:**

4

---

> ### Author Response · Authors · 2021-08-09
> **Authors' Response**
>
> We would like to thank the reviewer for the positive assessment and valuable feedback. We agree with the proposed corrections and will include them in the final version of the paper.

---

### Official Review · Reviewer_9AF3 · 2021-07-21

**Rating:** 7
**Confidence:** 4

**Summary:**

This work investigates the best-case lower bounds in the online learning setting -- meaning a lower bound on the regret value of an algorithm. Even though an algorithm is usually concerned with minimizing the regret and thus upper bounds on regret are used to describe algorithm performance, information about lower bounds on the regret can also provide information about the algorithm performance, its adaptivity, and in some cases certain notions of fairness. In this submission, the authors consider the FTRL family of algorithms, anytime (no prior knowledge of $T$) and timeless (first-order bounds), with constant, time-varying or adaptive learning rates. The results mainly show that the magnitude of regret of such algorithms stays within the range of the upper bound (or is nonnegative sometimes) with some exceptions. An application to online learning with group fairness is provided as a use case for such bounds.

**Limitations And Societal Impact:**

Societal impact was adequately addressed. Limitations of the work and possible future directions also discussed in a sufficient manner.

**Main Review:**

This work concerns itself with algorithm-specific regret lower bound in online learning, i.e. characterizing not only how low the regret of an algorithm is guaranteed to be but completing the range of it. It is true that the upper bounds are the important part of an algorithm's guarantee, they indicate how well the algorithm is sure to perform. These best-case lower bounds simply provide more information in regards to the algorithm's adaptivity -- a truly adaptive algorithm should be able to attain $-T$ regret for certain sequences. On the other hand, these results can be used in devising "fair" online algorithms, e.g. ones with group fairness as in this submission, in which case negative linear regret is not desirable at all (informally, one is aiming to balance mistakes out uniformly). Overall, this is work that can have sufficient significance in completing the community's knowledge about the online algorithms such as FTRL types that are used ubiquitously.

The paper is written clearly, is easy to follow, and the results are presented in an honest and straightforward fashion. The main criticism in my opinion can be its significance, but I believe such work can encourage more results in this direction and resolve most such questions in several years. Applications other than fairness might be born along the way too. I believe this submission is ready for publication and simply have some questions to the authors.

As shown, FTRL type algorithms can have $\geq 0$, $-\sqrt{T}$ and even $-T$ (linearized) regret while all have $\sqrt{T}$ regret upper bound. What is the end goal in characterizing such results then? Is the idea to exhaust all the algorithms that are around just to have the best-case lower bound results? Are there notions about what is desired?

Seems like general minimax bounds can't be shown since $-T$ is achievable, so what can be said in a general manner that doesn't depend on a specific algorithm? And these results still don't rule out the possibility of an unknown algorithm having desired upper bound *and* lower bound properties, so pessimistically seems like there is nothing substantial to say in here.

The authors emphasize that their analysis applies to anytime algorithms: just a question, if you have analysis for known horizon algs, then can't you similarly analyze such the algorithm of alg + doubling trick which is anytime.



**Time Spent Reviewing:**

8

---

> ### Author Response · Authors · 2021-08-09
> **Authors' Response**
>
> We would like to thank the reviewer for the valuable feedback and constructive criticism. We provide more detailed answers below.
>
> R: “As shown, FTRL type algorithms can have $\geq 0$, $-\sqrt{T}$ and even $-T$ (linearized) regret[..].“
>
> A: We actually do not have in mind a precise end goal for this research; we view this work as exploratory and hope that it leads to a deeper understanding of online learning. While this research pursuit did lead to some answers (for instance, in the fairness-related application), we ran into many new questions that we hope will spark future research activity. Below, we list some of the goals we have in mind.
>
> First, beyond the studied application to group fairness, we believe best case lower bounds (BCLB) provide another view on the adaptivity of online learning algorithms. In particular, when BCLB are much more negative than the upper bounds, and when the BCLB can be shown to be tight (perhaps via explicit constructions), this may indicate a strong adaptivity of the algorithm for particular instances. In such cases, further characterizing the instances which achieve these lower bounds seems like a fruitful research direction for further work.
>
> We also would like to stress a second goal here, in case it was not salient from the paper: while many algorithms have been developed that have been explicitly designed to have certain types of adaptivity, is it possible to provably rule out that other (presumably non-adaptive) algorithms truly cannot significantly outperform the best expert, even in the best case? Although not exactly related, such a question bears similarity to recent research of [Mourtada and Gaïffas (JMLR, 2019)](https://jmlr.org/papers/v20/18-869.html) showing that the standard Decreasing Hedge algorithm (for decision-theoretic online learning) automatically adapts to the lucky situation where each expert emits i.i.d. losses and there is a gap in performance between the mean of the best expert and the mean of the second-best expert. The same work also proved that Hedge with the doubling trick cannot enjoy this best-of-both-worlds behavior. We initially were curious whether the use of Hedge with a time-varying learning rate (like that used in Decreasing Hedge) could somehow allow the algorithm to, for certain sequences, significantly outperform the best expert. By showing that this can *never* happen, one obtains even stronger motivation to use algorithms that enjoy low shifting regret, when low shifting regret is what is desired. On the other hand, in applications where one wishes for an algorithm to not waver significantly from the performance of the best expert, our results give insight into what algorithms should be permitted.
>
> R: “Seems like general minimax bounds can’t be shown[..].”
>
> A: We agree that the definition of best-case lower bounds is algorithm dependent; yet, we believe they can still lead to more general treatment, in particular with connections to minimax rates as proposed by the reviewer. For example, the question of whether there exists a minimax optimal algorithm which also enjoys a $o(T)$ best-case lower bound (or a rate matching the upper bound) is interesting. We have shown this for OCO in some standard settings, but this question can be more generally posed. This relates to the recent trend of minimax rates under algorithmic constraints (e.g., limited computation, limited communication, or stability https://people.eecs.berkeley.edu/~wainwrig/Barcelona14/Wainwright_ICM14.pdf, https://arxiv.org/abs/1804.01619).
>
> R: “[..] Can’t you similarly analyze such algorithm of alg + doubling trick, which is anytime?”
>
> A: This is a keen observation, and we did think of this possibility. However, note that in the analysis based on use of the doubling trick, the algorithm actually competes with the best expert over each epoch (where epoch sizes double). Consequently, it is possible that, over each epoch, an algorithm has (say) $-\sqrt{T}$ regret, but as compared to the single best expert over the entire sequence of rounds, the regret is $-\Omega(T)$. How could this happen? Let the best expert in all but the last epoch be the same expert $j_1$, while the best expert over the last epoch is another expert $j_2$. Finally, assume that (considering the cumulative loss over the full game) that expert $j_1$ is linearly better than expert $j_2$, whereas in the last epoch expert $j_2$ is linearly better than expert $j_1$. In this situation, the doubling trick allows for sublinear regret with respect to the 1-shifting regret, and the best 1-shifting sequence of experts is linearly better than the best 0-shifting sequence of experts (i.e., the best constant sequence, also known as the best expert).

---

> > ### Comment · Reviewer_9AF3 · 2021-09-01
> > **Re: rebuttal**
> >
> > Thanks for the comments. I think this is an interesting paper that should be published at this venue, my score of 7 is unchanged.

---

### Decision · Program_Chairs · 2021-09-27

**Decision:**

Accept (Poster)

**Comment:**

Reviewers all liked the paper, which seems to provide an interesting new take on analysis of adaptive online learning.